# Physical modeling of ice-sheet-induced salt movements using the example of northern Germany

Jacob Hardt[1,2], Tim P. Dooley[2], Michael R. Hudec[2]

[1]Freie Universität Berlin, Department of Earth Sciences, Physical Geography, Berlin, Germany

[2]The University of Texas at Austin, Bureau of Economic Geology, Jackson School of Geosciences, Austin, Texas, USA.

*Correspondence to*: Jacob Hardt (jacob.hardt@fu-berlin.de)

**Abstract.** Salt structures and their surroundings can play an important role in the energy transition related to a number of
storage and energy applications. Thus, it is important to assess the current and future stability of salt bodies in their specific geological settings. We investigate the influence of ice-sheet loading and unloading on subsurface salt structures using physical models, based on the geological setting of northern Germany, which was repeatedly glaciated by the Scandinavian Ice Sheet during the Pleistocene. Apparent spatial correlations between subsurface salt structures in northern Germany and Weichselian ice marginal positions have been observed before and the topic is a matter of ongoing debate. Recently described
geomorphological features – termed surface cracks – have been interpreted as a direct result of ice sheet induced salt movement resulting in surface expansion. The spatial clustering and orientation of these surface cracks was so far not well understood, owing to only a limited number of available studies dealing with the related salt tectonic processes. Thus, we use four increasingly complex physical models to test the basic loading and unloading principle, to analyze flow patterns within the salt source-layer and within salt structures, and to examine the influence of the shape and orientation of salt structures with respect
to a lobate ice margin in a three-dimensional laboratory environment. Three salt structures of the northern German basin were selected as examples that were replicated in the laboratory. Salt structures were initially grown by differential loading, and buried before loading. The ice load was simulated by a weight that was temporarily placed on a portion of the surface of the models. The replicated salt structures were either completely covered by the load, partly covered by the load, or were situated outside the load extent. In all scenarios, a dynamic response of the system to the load could be observed: while the load was
applied, the structures outside the load margin started to rise, with a decreasing tendency with distance from the load margin and, at the same time, the structures under load subsided. After the load was removed, a flow reversal set in and previously loaded structures started to rise, whereas the structures outside the former load margin began to subside. The vertical displacements during the unloading stage were not as strong as during the load stage, and thus the system did not return to its pre-glaciation status. Modeled salt domes that were located at distance from the load margin showed a comparably weak
reaction. A more extreme response was shown by modeled salt pillows whose margins varied from sub- parallel to sub-perpendicular to the load margin, and were partly covered by the load. Under these conditions, the structures showed a strong reaction in terms of strain and vertical displacement. The observed strain patterns at the surface were influenced by the shape of the load margin and the shape of the salt structure at depth, resulting in complex deformation patterns. These physical

modeling results provide more evidence for a possible interplay between ice sheets and subsurface salt structures, highlighting
the significance of three-dimensional effects in dynamic geological settings. Our results lead to a better understanding of spatial
patterns of the surface cracks that were mapped at the surface above salt structures and offer further room for interpretation of
the influence of salt movements on the present-day landscape.

## 1 Introduction and rationale

Salt structures are unique geological features found in salt-bearing basins across the globe. At geological time scales, salt acts
like a fluid leading to the formation of a wide range of different salt structure types and highly complex deformation patterns
in the sediments surrounding the salt (Warren, 2016). In addition to the economic importance of salt for the chemical and
hydrocarbon industries, salt tectonics may play an important role in the energy transition (Duffy et al., 2023). Solution-mined
caverns inside salt bodies provide a vast storage potential for hydrogen (Lankof et al., 2022), and porous deformed sediments
around the salt provide storage potential for carbon dioxide (Zhang et al., 2022a; Zhang et al., 2022b). Salt caverns are also
targeted as potential repositories for radioactive wastes (Jackson and Hudec, 2017a). In addition, due to their relatively high
thermal conductivity, salt structures are of interest for geothermal energy production (Daniilidis and Herber, 2017; Moeck,
2014). For a safe and economic use of the outlined applications, it is crucial to understand salt tectonic processes at all scale
levels. For example, in the case of the storage possibilities, the long-term stability of the salt and its surroundings needs to be
reliably assessed.With regard to radioactive wastes, the long-term stability  needs to be predicted for up to 1 Ma, and future
glaciations are a factor that has to be considered in this case (BGE, 2020; Fischer et al., 2015). Only a precise understanding
of past salt tectonic processes will help us to predict future dynamics of salt-bearing basins, including but not limited to salt
movements and deformation of the surrounding sediments. These salt tectonic processes can be triggered by large scale tectonic
movements and changing sedimentary loads, which might be a result of changing climatic conditions including glaciations.
The load of the large Pleistocene ice-sheets pushed down the Earth's crust. In reaction to the unloading (i.e., melting of the
ice-sheets), glacial isostatic adjustment (Lambeck et al., 2014) and processes such as postglacial rebound (Spada, 2017) set in
and are still ongoing. As an example of postglacial rebound, Fennoscandia is still moving upwards, whereas regions to the
south of the Baltic Sea (such as the study area) are moving downwards (Bungum & Eldholm, 2022). In addition, the ice-sheet
advances modified subsurface hydro-thermal systems, which are still in the process of adapting to present-day conditions
(Amberg et al., 2022; Frick et al., 2022). In areas affected by glacial isostatic adjustment, observations from Europe (Brandes
et al., 2018; Sandersen and Jørgensen, 2022; Steffen et al., 2021; Štěpančíková et al., 2022) and North America (Adams, 1989;
Brooks and Adams, 2020; Bungum and Eldholm, 2022) demonstrated that tectonic fault zones can be reactivated, resulting in
glacially induced faults (Bungum and Eldholm, 2022; Steffen et al., 2014; Stewart et al., 2000). In northern Central Europe,
postglacial seismic activity has been identified at several preexisting faults (Brandes et al., 2015; Müller et al., 2021).
There is an ongoing debate on the impact of ice sheet oscillations on subsurface salt movements, which was initiated by
geomorphological observations. Spatial correlations between ice sheet extents and subsurface Zechstein salt structures in

northern Germany have been reported since the 1950s (Gripp, 1952; Schirrmeister, 1998). Furthermore, the influence of subsurface salt structures on the modern river network and the general topography of northern Germany has been discussed (Sirocko et al., 2008; Sirocko et al., 2002; Stackebrandt, 2005). White (1992) discussed the spatial relation between the extent of the Laurentide Ice Sheet of North America and the northern major salt bodies, highlighting that the end moraines are

"concentric with isopachs of the salt" (White, 1992: 313). Theoretical models have evolved to explain salt movements triggered by loading and unloading processes of ice sheet advances, which in return influence flow paths of the ice sheets (Liszkowski, 1993; Sirocko et al., 2008). In a recent study, Hardt et al. (2021) described distinct geomorphological features ("surface cracks") in the young morainic area of northern Germany, which show a spatial correlation to subsurface Zechstein salt structures, and which are interpreted as surface expansion ruptures due to ice-sheet induced salt movements. Despite the

occurrence of the surface cracks above some salt structures, questions remain regarding the spatial clustering, the orientation of the surface cracks, and why they are only found above some salt structures and not all of them (Fig. 1).

Lang et al. (2014) used numerical 2D models to simulate loading- and unloading effects of an ice sheet on salt structures (salt domes and salt walls) and the associated possible deformations (Lang and Hampel, 2023). As a result, downward displacement of the salt structures during the loading phase and an upward displacement during the unloading phase could be confirmed by

the models. However, the derived absolute amounts of displacement were relatively low, ranging from -37m to +4m (Lang et al., 2014) and -11 to +3.6m (Lang and Hampel, 2023), depending on the model parameters. Thus, although the physical principle was confirmed by Lang et al. (2014) and Lang and Hampel (2023), the authors questioned a significant influence of ice-sheet induced salt movements on the surface topography, in contrast to previous findings by other authors (Hardt et al., 2021; Sirocko et al., 2008; Sirocko et al., 2002; Stackebrandt, 2005).

However, these previous model approaches suffer from two shortcomings. First, previous workers never modeled what would happen if the edge of an ice sheet was emplaced halfway across a buried salt structure, as is believed to have happened in several places in northern Germany. Second, 3D effects of salt flow associated with a complex, lobate ice front were ignored. Some important questions that persist are:

1. What happens if salt structures are only partly covered by the ice and which role does the type of salt structure play?
This requires an investigation of different shapes and sizes of the salt structures during loading- and unloading processes.

2. What influence do the geometries of both ice margin and subsurface salt structure have on salt flow patterns? This requires us to investigate loading- and unloading-induced intrasalt flow patterns.

3. Can these results help us to understand spatial patterns of present-day geomorphological features, such as surface
cracks, above salt structures in northern Germany?

In order to answer these questions, we built a series of scaled physical models, which allowed us to visualize and quantify the 4D evolution of salt structures subjected to loading- and unloading processes under controlled laboratory conditions.

## 2 Study area

Northern Germany constitutes an ideal study area, as it is rich in various types of subsurface salt structures, was repeatedly
glaciated during the Pleistocene, and provides several areas where geomorphological landforms point to a salt tectonic influence.

Located within the Southern Permian Basin of the Central European Basin System (CEBS), Zechstein salt is abundant in the subsurface of northern Germany (Warren et al., 2008; Warsitzka et al., 2019; Fig. 1). A combination of sedimentary loading and tectonic forces triggered the evolution of numerous salt structures (Scheck-Wenderoth et al., 2008; Scheck et al., 2003;
Strozyk et al., 2017), varying in shape and dimension from elongated pillows to walls and domes (Stackebrandt and Beer, 2015). The main structure types in the NE German sector of the CEBS are salt pillows, although several domes exist, too. Salt pillows have a parallel contact with suprasalt strata, whereas salt domes have discordant contacts with their upper strata (Jackson & Hudec, 2017b). The Mesozoic and Cenozoic overburden on the Zechstein salt varies in thickness in the region between more than 3000 m above deep-seated pillows, to only few hundred m above the highest salt domes (Stackebrandt and
Beer, 2015) – some domes in northern Germany even pierce to the land surface (Künze et al., 2013; Sirocko et al., 2002; Stackebrandt, 2005).

The onset of salt rise in the study area was probably triggered by tectonic extension in the Middle to Late Triassic (Scheck-Wenderoth et al., 2008). Intense tectonic deformation during the Late Jurassic and Early Cretaceous, as well as tectonic compression during the Late Cretaceous to Early Cenozoic resulted in further main phases of salt movement (Warsitzka et al.,
2018). Since the Late Paleogene, tectonic stresses and salt tectonic movements have decreased (Strozyk et al., 2017). Although Pleistocene ice-sheet loading and glacial isostatic adjustment is seen as a potential trigger for younger salt movements (Reicherter et al., 2005), information on this in the study area is sparse and related only to few single salt structures (e.g., Sperenberg, Rambow, Rüdersdorf; Ludwig & Stackebrand, 2010). In general, it is challenging to differentiate ice-sheet induced salt tectonic movements from other (longer-term) tectonic movements, as the salt structures are usually coupled to
tectonic lineaments (see discussion in Hardt et al., 2021). Therefore, approaches that take into account both the geomorphology and the deeper subsurface are necessary.

During the Weichselian, the study area was transgressed by the W1 advance, which occurred in late Maritime Isotope Stage 3. The Weichselian W2 advance occurred in Maritime Isotope Stage 2 and corresponds to the Last Glacial Maximum. The W2 advance reached only into the northern parts of the study area (Fig. 1; Lüthgens et al., 2020).

Localized landforms directly related to ice-sheet loading induced deeper salt movements were documented in the study area by Hardt et al., 2021. The so-called surface cracks are interpreted as expansion ruptures due to salt flow triggered by loading-und unloading effects of the SIS , which eventually resulted in upwards movement of pillows and domes. The described landforms are up to several km long, can be up to 20 m deep and are several tens of m wide (Fig. 2, 3, 4).

Interestingly, these features are found within the extents of the Weichselian ice sheet advances and cluster especially around
the ice marginal position of the W2 (LGM) advance. At some of the structures where surface cracks are present, salt tectonic

movements during the Pleistocene had been discussed before in geological contexts, e.g., at the Sperenberg salt dome (Stackebrandt, 2005), the Stolpe salt pillow (Böse, 1989) or at the salt pillows beneath the W2 terminal moraines (Gripp, 1952; Schirrmeister, 1998).We took three natural prototypes of salt pillows and domes from northern Germany where surface cracks are present (Fig. 2, 3, 4) as guides for our experimental setup detailed in the following section.

## 3 Experimental methodology and setup

### 3.1 Modeling materials and data capture

We used material commonly used in the physical modeling of salt tectonics: ductile silicone represented the rock salt and dry, brittle, sand-sized material represented the sedimentary overburden (e.g., Dooley et al., 2009). The polydimethylsiloxane silicone had a density of 950 to 980 kg m-3 with near-Newtonian viscous characteristics. Powdered pigments were mixed with the silicone and added as passive markers to several locations in the source layer in order to track the silicone flow. The sedimentary overburden was represented by differently colored layers of dry sand-sized silica grains with grain sizes ranging between 300-600 μm and a bulk density of ca. 1700 kg m-3. These sands were mixed with hollow ceramic microspheres with a grain size ranging between 90 to 350 μm and a bulk density of 650 kg m-3 in order to moderate the bulk density of the overburden load. These microspheres were also added atop of the model "salt structures" (except for run 1 only above the pillow structure, see details below), and the resulting reduction of the overburden bulk density above the structures facilitated their growth during the first stage, as the denser materials surrounding the model salt structure created a differential load. The topmost sand layer was an especially brittle material with a mixture of finer-grained microspheres ($\leq$ 80 μm) and our finer silica sands ($\leq$ 300 μm), which was intended to better display any structures that formed at the surface. See Dooley et al. (2009), Dooley and Schreurs (2012), and Reber et al. (2020) for more detailed information on the physical properties of the selected modeling materials.

The surfaces of the models were photographed at fixed time intervals by computer-controlled cameras. The digital image correlation (DIC) system comprised a high-resolution stereo charge-coupled device (CCD) system (see (Adam et al., 2005). This setup allows for tracking the surface strain, as well as recording vertical and horizontal displacement rates at a sub-mm scale ("Z maps"; Fig. 5). As the base of the model was a transparent acrylic plate, the movements of the pigmented spots inside the silicone layer were recorded by time-lapse photographs from an additional camera mounted on the floor underneath the model at the same interval as the stereo system. These flow markers only provide a partial history of flow in the source layer as their distribution is spatially limited. After each finished run, the models were impregnated with a gelatin solution (left to dry for at least 12 h), which enabled slicing the models into thin (ca. 0.5 cm) slabs. Photographs of each slab allow for an interpretation of subsurface fault patterns.

## 3.2 Model design

For the sake of comparability with natural structures, we chose the shapes of three salt structures from northern Germany, the Groß Schönebeck (GS; Fig. 2) and Triepkendorf-Klaushagen-Flieth (KH; Fig.3) salt pillows, and the Netzeband (NB; Fig. 4) salt dome (Stackebrandt and Beer, 2015). The GS and KH salt pillows have an undulating topography with several peaks. The GS pillow is oriented perpendicular to the ice margin, whereas the KH pillow is grossly parallel to the W2 ice marginal position (Fig. 1; Fig. 4). W2 end moraines have been mapped in the immediate surroundings of the KH pillow and right at the surface above it. To the northeast of the KH pillow, prominent push moraine complexes are visible in the digital elevation model (Fig. 3), which are probably recessional moraines of the W2 advance (the so-called Angermünder Staffel; (Liedtke, 1981). These three structures were selected because many very clear surface cracks were detected at the land surface above them (Fig. 2, 3, 4). In addition, both pillows were partly transgressed by the gLGM ice advances, which provides a setting that corresponds to the focus of our research questions (Table 1; Fig. 2 + 4). Deeper subsurface features beneath the salt structures were not included in the models.

We decided to start with a rather simple model setup in order to see whether the basic loading- unloading mechanism can be reproduced (runs 1 and 2). More complexity was added after this first goal was achieved (runs 3 and 4).

The models were scaled so that 1 cm in the model represents roughly 1 km in nature (a scaling factor of $10^{-5}$). The base of all models was a ca. 0.8 cm thick layer of ductile silicone representing the Zechstein salt source layer. On top of the silicone, layers of silica sands were added to simulate the overburden. In order to recreate the distinct salt structure shapes from the study region, styrofoam templates were placed over the silicone layer while the first sand layer with a density 1.4 times that of our salt analog was added to the model. The styrofoam templates were then removed, and hollow microspheres infilled the voids left by the templates. These cenospheres are less dense than both our sand layer and our salt analog. This setup imparts a differential load onto the salt analog causing rise in the areas with a lower-density cover. All subsequent layers of sand were added across the entire model without compacting them, just cresting our rising pillows and diapirs. In this way the model surface was flattened after each load was applied.

In order to simulate the ice load, a thin metal plate was placed on one side of the sandbox. In the first two runs, that metal plate was perfectly rectangular, resulting in a straight load margin (Fig. 5). In the later runs, a metal plate with an undulated front edge was used to simulate the lobate nature of ice margins (Fig. 5). The undulated shape was inspired by the course of ice marginal positions atop the investigated salt structures in northern Germany, which were reconstructed from published maps (Liedtke, 1981) and from analysis of the digital terrain model (GeoBasis-DE/LGB, 2020). The plates served to evenly distribute the load of the weights that were placed on top of them. The weights were partly covered with granular materials so that also the loaded side of the experiment could be tracked with the camera system.

All model runs went through the same three stages (Fig. 5):

- Stage 1 (initial growth stage, ca. 72 h): Preparation of the model environment before start of the actual experiments. Differential loading of the salt analog, causing rise of the silicone in the regions of our template shapes to create

distinct structures with outlines similar to those seen in the study area. In this stage the salt analog reacts solely to gravity and differential loading. Stage 1 finished when overburden reached a sufficient thickness and when structures became mostly inactive (i.e., ceased to rise).

- Stage 2 (load stage, ca. 48 h): A metal plate was added to the surface of one portion of the model, which was weighted down with additional weights to simulate the ice sheet load.
  - In run 4, stage 2 was split in stage 2 A (loading for 24 h) and stage 2 B (partial unloading for 24 h, see Fig. 5)
- Stage 3 (unloading stage, ca. 48 h): The metal plate and the weights were removed. Models were monitored until no more active movement was detected (at least 48 h) and then solidified with a gelatin solution and serially sectioned.

## 3.3 Remarks on the selection of model parameters

In order to setup a physical model that is able to answer our questions detailed above, the key parameters to be modelled need to be identified and selected. It is in the nature of a model that not all involved parameters can be implemented, and it is possible that not all of the key parameters have in fact been identified yet. While a scaled model allows us to compare and contrast observed processes and features with phenomena discovered in nature, a factor such as time is difficult to scale from geological dimensions to dimensions measurable and practically workable in the laboratory (Jackson and Hudec, 2017a). Thus, we decided not to implement the time-based effect of an advancing ice sheet that incrementally covers larger parts of a certain area and did not try to simulate the downwasting phase of the ice sheet over a certain time span in detail. In the realm of the geological time scale, however, the advance and disappearance of ice-sheets during a few thousand years (Hughes et al., 2016), can be considered more or less instantaneous. Thus, both, the loading and unloading occurred instantly in the model in most of the runs shown here. Only in run 4 was the load stage split into two sub-stages. In the first sub-stage, the full load extent was applied, and in the second sub-stage a part of this area was unloaded. As the goal was to create a setting that can be compared to the northern German lowlands, we concentrated on approximating the vertical and horizontal dimensions of selected representative salt structures, the thickness of the sedimentary overburden, and the ice-sheet coverage, which only affected a part of the area. The dimension of the ice sheet pressure (i.e., the absolute load weight), however, was not dynamically scaled in our experiments.

We explicitly did not scale the applied weight of the load and decided to use a relatively heavy load (20 - 30 lbs.; 9 - 13.6 kg; Table 2) to begin with. While the dimensions of displacement and strain recorded in the laboratory may be exaggerated compared to natural conditions, we primarily aimed to understand the basic principles and flow dynamics induced by an (ice) load on subsurface salt. The advantage of the relatively heavy weights that we used was that we triggered an immediate and significant reaction of the salt system. We are confident that lower weights in the end would have produced comparable results (albeit at a smaller order of magnitude), with the disadvantage that more laboratory time would have been necessary and fewer experiments could have been carried out. However, under these experimental conditions we refrain from converting our results

into natural dimensions (e.g., absolute vertical displacement in m), as our primary goal was identifying salt tectonic processes, flow vectors within the salt, and comparison of the results to our natural prototype.

## 4 Results

### 4.1 Half loaded pillow, two domes, straight load margin

The first two runs used a similar setup with two domes inspired by the shape of the Netzeband (NB) salt dome and a pillow structure resembling the Groß Schönebeck (GS) salt pillow. The results from both runs were broadly similar, and for brevity, we will mainly focus on the results of run 2 in the following paragraphs. The differential loading approach successfully favored the development of the structures following the predesigned shapes. Also for reasons of brevity, the results of the growth stage of the experiments will not be further presented here, as they do not contribute significantly to answering the research questions.

While one of the domes was placed in the northern section of the model and was loaded, the second dome was placed at some distance south of the load margin and was consequently never covered by the load. In case of the GS pillow, only the northern half was covered by the load, as was the case during the W2 ice advance in our study area.

The application of the load (stage 2) to the northern side of the model triggered an immediate and sustained reaction of the system, with a decreasing intensity over the course of 48 h. While the loaded area subsided in response to the load, the unloaded

areas – especially above the salt structures – started to rise (Fig. 6 A). The vertical displacement was the highest above the unloaded part of the GS salt pillow, from where it extended the most toward its southeastern flank. The vertical displacement above the southern NB salt dome was only about half as strong and generally not much stronger than in the adjacent areas between and around both structures.

An arched crestal graben structure containing several longitudinal extensional surface fractures formed above the GS pillow,

striking parallel to the long axis of the buried salt structure. The strain data shows that the highest extensional strains occurred spatially focused close to the load margin at the eastern flank of the GS pillow (i.e., the onset of the graben structure; white arrow in Fig. 6 B).

After the removal of the load (stage 3), the Z-map shows that the formerly load-induced highs in the unloaded portion of the model started to subside, and that some of the formerly loaded areas started to rise (Fig. 6 C). In both cases, vertical

displacement rates above and in the vicinities of the GS structures were the highest. However, the amount of vertical displacement amounts to only about 50 % (Table 3) compared to the vertical movements induced by the load (stage 2, Fig. 6 A). The arched crestal strain pattern above the GS structure that developed in stage 2, continued to develop also across parts of the northern (formerly loaded) section of the structure (white arrows in Fig. 6 D). Again, the deformation was less than half as intense as in stage 2 (Fig. 6 D). The Z-maps show how the whole area close to the load margin reacts to the loading and

unloading, and the NB domes extend this area a bit to the north and south. However, given the overall less intense reaction to

the unloading, the measurable displacement rates above the northern dome accounted for only ca. 30 % of those measured above the northern GS pillow.

Flow patterns recorded by passive markers within the salt analog varied drastically during the three different stages, and the patterns are consistent with the results from the Z-maps and strain data. During stage 1, the flow of the colored marker dots was directed towards the nearest salt structures, at a comparably low pace (green arrows in Fig. 7). With the onset of stage two, as the load was added to the northern half of the model, the pace increased and a more or less southwards oriented trend (at right angle to the load margin) was established (blue arrows in Fig. 7). The salt bodies to the south of the load margin pulled in the salt from the source layer, facilitating their rise. After unloading in stage 3, a reversal of the flow direction could be recognized, especially in the not loaded area and close to the load margin. However, the pace and intensity of flow was considerably smaller than in stage 2 (red arrows in Fig. 7).

The sections provide some additional insights into internal deformation structures. All sections represent the status after the last (unloading) stage. In runs 1 and 2, normal faults originating from the top of the salt analog bound the crestal graben structures that follow the long axis of the GS pillow structure (Fig. 8; run 1, section 53; run 2, section 45). In contrast, the comparably small vertical displacement of the NB domes was not accompanied by the development of discernable faults, (Fig. 8; run 1, section 31; run 1, section 68).

### 4.2 Two pillows, curved load margin

In runs 3 and 4, two differently oriented salt pillows were modelled. Whereas the GS pillow was more or less orthogonal to the load margin, the newly added Klaushagen (KH) pillow was broadly parallel to the load margin (Fig. 9). In run 3, the load margin was curved and resembled the course of the W2 ice marginal position. As in the previous runs, an arched graben structure extending southwards from the load margin formed above the GS pillow. Above the KH pillow, which was oriented broadly parallel to the undulating load margin, arched graben structures formed that followed the shape of the underlying structure. Thus, due to the curved shape of the KH pillow and the differently curved shape of the load margin, a multitude of convergent stress vectors seem to have developed in relation to the KH pillow, which become apparent in the strain maps (Fig. 9). The outcome of this was surface salt extrusions that occurred above the KH pillow in runs 3 and 4, even though the weight of the load had been reduced from run 3 to run 4 in order to limit this process. Above the GS pillow, an extrusion occurred only under the higher load conditions of run 3.

In model run 4, we used an even more undulating load margin as in run 3 for the first 24 h and then removed part of that load for the next 24 h, which is located between the W2 main ice marginal position and several recessional ice marginal positions (see Fig. 5 for illustration). By doing so, we aimed at modeling a more dynamic load distribution pattern during the downwasting phase. Especially noteworthy is that the load margin above the GS pillow was rather straight in run 3, whereas it was curved and followed an NE-SW direction in run 4.

The partial unloading in stage 2 B of run 4 resulted in uplift of the previously loaded area and slight subsidence of the region outside the former load margin. The absolute amounts of vertical displacement were significantly smaller than those of stage

2 A (max. 4 mm vs. max. 10 mm; Fig. 10 Stage 2 B Z). As in stage 2 of run 3, a strain pattern following the long axis of the KH pillow had developed (Fig. 10 B).

The flow patterns of runs 3 and 4 reflect the increased complexity of the model design, as compared to runs 1 and 2. During the loading in stage 2 A of run 4, the flow markers divert away from the loaded zone (Fig. 11). Flow markers outside the loaded area move relatively straight away from the margin. In the loaded area, the influence of the salt pillows on the flow dynamics becomes visible. A marker in the northeast split into two and fed into both pillows. The marker west of it also split, with one part feeding into the eastern tail of the KH pillow and the other half feeding into its central part. The most intense movements were recorded in the western "corner" of the load margin, where an extrusion occurred in the course of the stage. During the partial unloading in stage 2 B, the general flow directions of the markers maintained the same, albeit at a much lower pace. A reversal of the flow direction apparently set in only after the full unloading in stage 3 (Fig. 11).

The change in the ice margin shape above the GS pillow in run 4 allows us to analyze the influence of this geometry on the resulting strain patterns. In runs 1, 2, and 3 the load margin above the GS pillow was linear across this buried salt structure, whereas in run 4 it was curved in order to imitate a lobate ice tongue. In the first three runs, the strain patterns above the pillow were very similar at the end of stage 2. The surface was arched upwards and a crestal graben structure was following the long axis of the buried salt structure (white arrows in Fig. 12 A1 + A2). In run 4, where the load margin crossed the structure diagonally (Fig 12 B1 + B2), the pattern deviated from that scheme. Here, the crestal graben structure following the long axis of the structure, had not developed as clearly as in the previous runs. The deformation in this case was in general focused closer and in parallel to the load margin (white arrows in Fig. 12 B1 + B2).

In runs 3 and 4, a fault reaching from the source layer up to the surface had developed at the load margin (Fig. 8; run 3, section 44). In run 3, the fault occurred near the inflection point of the lobate edge of the load (ca. -80 cm E / 100 cm N in Fig. 9). The high load conditions of run 3, which caused extrusions at the surface, resulted in the complete expulsion (welding) of an area of the source layer into the not loaded structures north of the GS pillow in (Fig. 8; run 3, section 97).

## 5 Discussion

Here, we will answer the questions outlined in the introduction of our paper and discuss our results in the context of previous studies.

*What happens if salt structures are only partly covered by the ice and which role does the type and size of salt structures play?* The areas with the highest detected uplift rates and deformation are located above the partially loaded pillow structures. This is indicated by the associated strain patterns parallel to the long axes of the structures and the load margins. When a salt structure is partially loaded, there is a high connectivity between loaded and unloaded parts of the structure, so large volumes of salt can move from one part of the structure to another. Salt structures outside the load margin lack this intrinsic link and

thus show a weaker reaction than the partially loaded ones. As a result, partial loading of salt structures appears to be one of the key factors in the interplay between ice-sheet loading and salt remobilization.

When comparing our different modeled structures in plan view, it is apparent that the bigger structures (in this case the pillows) showed significantly stronger reactions to the loading and unloading cycles than the smaller structures (in this case the domes), irrespective of roof thickness and strength. This is partially in agreement with the observations from northern Germany, where

the most intense surface crack clusters are found at the surface above some deeply buried salt pillows, and not above the shallower salt domes. This may appear counterintuitive, as one might expect more deformation in relation to the shallowly buried salt structures, because their roof is weaker.

As outlined before, a crucial point related to this question is the intrinsic connectivity of the partly loaded salt bodies. Given the bigger planforms of the salt pillows, as compared to the salt domes, and their widespread distribution in northern Germany

(Fig. 1), these deep pillows have a greater potential for being partly loaded by an ice sheet than the domes. All other things being equal, we would assume more deformation out of the shallowly buried salt structures, due to their weaker roof.

We followed the natural examples from northern Germany, where salt domes are not located in the close vicinity of ice marginal positions (Fig. 1), and consequently the NB domes were placed at a distance from the load margin in our models. These structures lacked connectivity as they were not part of a salt corridor at depth and were only fed via the thinned source

layer, which was additionally confined by the rim synclines adjacent to the diapirs, which increase the resistance of salt to flow (Hudec & Jackson, 2007; Wagner & Jackson, 2011). We assume that if they were placed closer to the load margin, or partially loaded, larger displacement rates would have to be expected.

*Which influence do the geometries of both ice margin and subsurface salt structure have on salt flow patterns?*

When combining a curved load margin with a complex partly loaded pillow shape, differently oriented stress vectors seem to overlap, resulting in a variety of strain, as well as uplift, patterns. This highlights the significance of three-dimensional effects between processes at the surface and the morphology of the structures at depth. This becomes especially evident when comparing a straight and a lobate load margin atop the GS structure (Fig. 12), where one vector is broadly parallel to the load margin and another vector follows the long axis of the structure at depth. In addition, the results indicate that converging salt

flow can be favored under specific configurations of load margin shape and geometry of the salt structure, which is again a result of three-dimensional stress propagation and the resulting response of the salt system. The strongest salt movements were recorded in runs 3 and 4, where salt extrusions occurred at the surface (Fig. 9 & Fig. 10). Here, the pillows were partially loaded (high connectivity) and the load margin was curved. Especially around the inflection points of the load margin, the flow lines concentrated. This is additionally supported by the extrusion in run 4, where extrusion occurred despite the reduced

absolute amount of load.

*Can these models help us to understand spatial patterns of present-day geomorphological features, such as surface cracks, above salt structures in northern Germany?*

Keeping the spatial extents of the two different Weichselian ice advances in mind (Fig. 1), the distribution of the surface cracks
may be explained on basis of the results gained from our physical models. From the natural observations in northern Germany, it is now obvious that the surface cracks are unevenly distributed across the salt structures. The surface crack clusters with the highest amounts of cracks are found along the W2 ice marginal position, where several large salt pillows are parallel or at an angle (in case of Groß Schönebeck/GS) to the W2 terminal moraines (Fig. 1; Hardt et al. 2021). Interestingly, the spatial correlation between salt pillows and the W2 ice marginal position has previously led to the development of the theory of a
dynamic relationship between salt structures and the ice extent (Gripp, 1952; Schirrmeister, 1998) and our results revealed the largest deformations in comparable settings. Although there are also lots of other salt bodies farther to the south in the hinterland of the older W1 ice advance, the crack clusters are not as well developed as in the vicinity of the W2 ice extent. The physical model results show that the distance to the load margin and the relation between the geometry of the load margin and the geometry of the (half-)loaded salt structures play a decisive role in the intensity and distribution of the load-induced salt
movements. The salt pillows along the W2 ice marginal position were partially loaded by a stagnant ice margin – this scenario yielded the largest displacements in our laboratory experiments. The salt inside the loaded areas of the pillows flowed toward their not-loaded counterparts (high connectivity). Thus, it is in line with our results that some of the most intense crack clusters detected so far were found atop the KH and GS salt pillows in Brandenburg. There are, however, other pillow structures along the W2 ice marginal position, where no surface cracks have been mapped yet. This highlights the need for further research
into the possible surface expressions of young salt movements.

An example from the field where the model results help to understand the spatial patterns of the surface cracks is the Thomsdorf cluster atop the KH salt pillow (Fig. 3 A). Here, some exceptionally deep (> 20 m) and long (up to 6 km) surface cracks were mapped and display two different primary orientations (Hardt et al., 2021). The orientation of the cracks of the northern part of this cluster are more or less in parallel to the W2 ice marginal position, whereas the cracks in the southern part of this cluster
are sub-perpendicular to the ice marginal position. These different orientations reflect different stress vectors, which was also observed in the laboratory when using a curved load margin above pillow structures: one strain pattern is parallel to the curved load margin, whereas the other strain pattern is oblique to the load margin following the long axis of the salt structure. Both patterns can be seen in the orientations of the Thomsdorf cluster (Fig. 13).

An interesting question remains as to whether reactivated salt structures played a role in defining the maximum extent of the
W2 advance. Several authors suggested that rising salt structures in front the advancing ice sheets may form physical barriers that obstruct, or at least impede, further ice flow (Schirrmeister, 1998; Sirocko et al., 2008)..Lang et al. (2014) concluded that salt rise alone would not be sufficient to create obstacles large enough to stop an inland ice sheet, but suggested that rising salt structures in the foreland of an advancing ice-sheet may favor the formation of glacitectonic thrusts. We did not model an incrementally advancing ice sheet, but our results so far corroborate the assumptions that ice load-induced salt rise may form
local surface obstacles to continued ice advance as highly connective structures such as the big pillows in our models are inflated by displaced salt. Salt flow is clearly directed away from the load margin. However, further research is necessary to better understand the relationship between ice marginal positions and salt structures.

*Comparison to numerical modeling studies*

In the 2D numerical models of Lang et al. (2014) and Lang & Hampel (2023), the salt domes (or in one example the salt wall) were either completely transgressed by the ice sheet, or the ice marginal position was located 1000 m "north" of the salt domes. The resulting vertical displacement rates were overall rather low (ranging between −37 up to +4 m). Looking at the basic processes, these results are more or less in agreement with our results on the dome structures: The loaded domes subsided during the load phase and showed very little upwards movement after the unloading. The domes outside the ice extent showed

a bit more upwards movement. As stated before, we assume that the results might differ if the domes were closer to the load margin. However, the scenarios that resulted in the most significant displacement rates in our physical models, i.e., partly loaded irregularly-shaped pillows, were not implemented in the numerical models documented in Lang et al. (2014) and Lang & Hampel (2023). Our model results show that three-dimensional effects between both load margin and salt structure geometry are crucial factors that need to be included in the analyses. In addition to the shape, also the size of the salt structure matters:

the bigger the structure the bigger the potential volume of salt that can be expelled from the loaded parts of the structure. Based on their model results, Lang & Hampel (2023) questioned the impact of ice sheet induced salt movements on the landscape. Although our experiments do not allow for quantification of absolute vertical displacement rates, they do provide evidence that it is within the realm of possibility that under certain circumstances the ice-salt-interplay does impact Earth surface processes, producing measurable and mappable surface deformation structures as documented by Hardt et al. (2021).


*Reflections on the scaling of our models*

As outlined in the methods section, we decided not to scale the ice load for this set of experiments, as we were mainly interested in understanding three-dimensional processes rather than absolute rates. For practical reasons, we used a very high load, as we expected this to provide an immediate reaction of the system, which gave us the time to experiment with other parameters such

as the salt structure types and the shape of the load margin. In addition, a dynamically scaled load would have introduced additional sources of uncertainty into the experiments, which would require choosing the ice-sheet thickness, the temporal dimension of the loading, or advance- and retreat rates.

As a result, the vertical displacements recorded in our models were very high, which is also reflected by the salt extrusions that occurred in runs 3 and 4. From the physical modeling perspective of salt tectonics, vertical displacements are usually high

where precursor diapirs and/or pillows are present, as these precursor structures focus deformation (Dooley et al., 2007, 2009; Dooley et al., 2015; Duffy et al., 2018). Despite the unrealistically high absolute vertical displacements, the insights into the three-dimensional flow behavior of the salt was nevertheless very important and paves the way for further research.

Although we witnessed a flow reversal during the unloading stages, the vertical displacement rates during the unloading stages only accounted for roughly 50% compared to those from the loading stages. This is most likely an effect of the very different

body forces involved, which were high due to the high applied load during the load stage and low (only gravity and sedimentary overburden) during the unloading stage. In addition, the back flow was favored within the pillows, where the salt is thick and

connectivity is high. Outside the pillows, the source-layer salt was thinned during loading by expulsion into the pillows and diapirs, and thus the flow resistance through these thinner conduits increased for the unloading stage. This thinning and increased flow resistance impacted the process of reequilibration driven solely by gravity, which we would expect to have
continued very slowly for some time to come. The process of decreasing salt flow in thinning salt layers is well known from research into salt welds and has to do with specific salt viscosities and internal impurities within the salts (e.g. Wagner & Jackson, 2011; Jackson and Hudec, 2017a).

## 6 Conclusions

We successfully implemented a series of physical models to simulate the effects of ice-sheet loading and unloading on
subsurface salt movements. Although we did not scale the absolute amount of the ice-sheet load, our results are in partial agreement with previous numerical studies and yield some important insights in salt tectonic processes beyond previously published data.

As a general observation from all model runs, the application of a load to a part of the surface, as well as the subsequent unloading, induce movements of the buried salt analog within the pillows and domes and in the intervening areas within the
source layer. These movements result in subsidence of the loaded areas and uplift of certain parts of the not loaded areas. Subsequent unloading results in uplift of parts of the formerly loaded regions and in subsidence of some areas outside the extent of the former load. The reversed vertical displacement after the unloading, caused by flow reversal of the salt system, accounts for only up to roughly 50 % of the vertical displacement that occurred during the loading stage.

In terms of strain and vertical displacement, the strongest movements occurred outside and close to the load margin and on top
or in close vicinity to partially loaded pillow structures. When a salt structure is partially loaded, there is a high intrinsic connectivity between loaded and unloaded parts of the structure, so large volumes of salt can move from one part of the structure to another. The physical models showed that the orientation of surface strain is controlled by the shape of the load margin and the contour of the pillow structure. In complex situations where both are curved, complex convergent flow patterns are generated by overlapping strain vectors result in differently oriented surface features, which are usually parallel either to
the load margin or the pillow structure. Additionally, where the geometry of the load margins and the pillow outlines favor converging flow, the highest displacement rates occurred. Our results emphasize the significance of three-dimensional stress propagation and salt flow, which need to be considered in the context of modeling studies investigating the interplay between surface processes and buried salt structures.

Many of the deepest and longest surface cracks in northern Germany have been mapped at the surface above salt pillows that
were partially loaded during the W2 maximum extent in Brandenburg. Our model results help to understand the different orientations of the surface cracks by overlapping stress vectors, which were introduced by the lobate shape of the ice margin, and the topography and outline of the salt structures at depth.

Despite being buried at great depth, salt structures can react to changing climatic conditions (i.e., glaciations) and have the potential to trigger Earth surface processes. Continued research is necessary to further understand the impact of ice sheet

induced salt movements on the landscape beyond the surface cracks. Considering the relevance of salt structures in the energy transition, it is crucial to gain a more detailed knowledge on surface-subsurface couplings and the related salt mobilization at depth. Geomorphological analyses at the surface above salt structures may be an additional method helping to assess the suitability of certain structures for storage or other applications.

## Code/Data availability

Video files of the physical model runs are available at TIB AV Portal: https://doi.org/10.5446/63073

## Author contributions

JH: conceptualization, visualization, funding acquisition, formal analysis, investigation, writing – original draft preparation, writing – review & editing. TD: conceptualization, methodology, investigation, visualization, model analyses, resources, software, writing – review & editing. MH: conceptualization, resources, writing – review & editing.

## Competing interests

The authors declare that they have no conflict of interest.

## Acknowledgements

This work was supported by a postdoc fellowship of the German Academic Exchange Service (DAAD) granted to JH. JH wants to thank all colleagues at the Applied Geodynamics Laboratory (Bureau of Economic Geology, The University of Texas

at Austin), especially Frank Peel, Oliver Duffy, and Kurt Tollestrup, for the many fruitful discussions on salt tectonics related processes and hospitality. We thank our reviewers Jörg Lang, Chris Clark, and Peter B. E. Sandersen for their constructive comments.

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

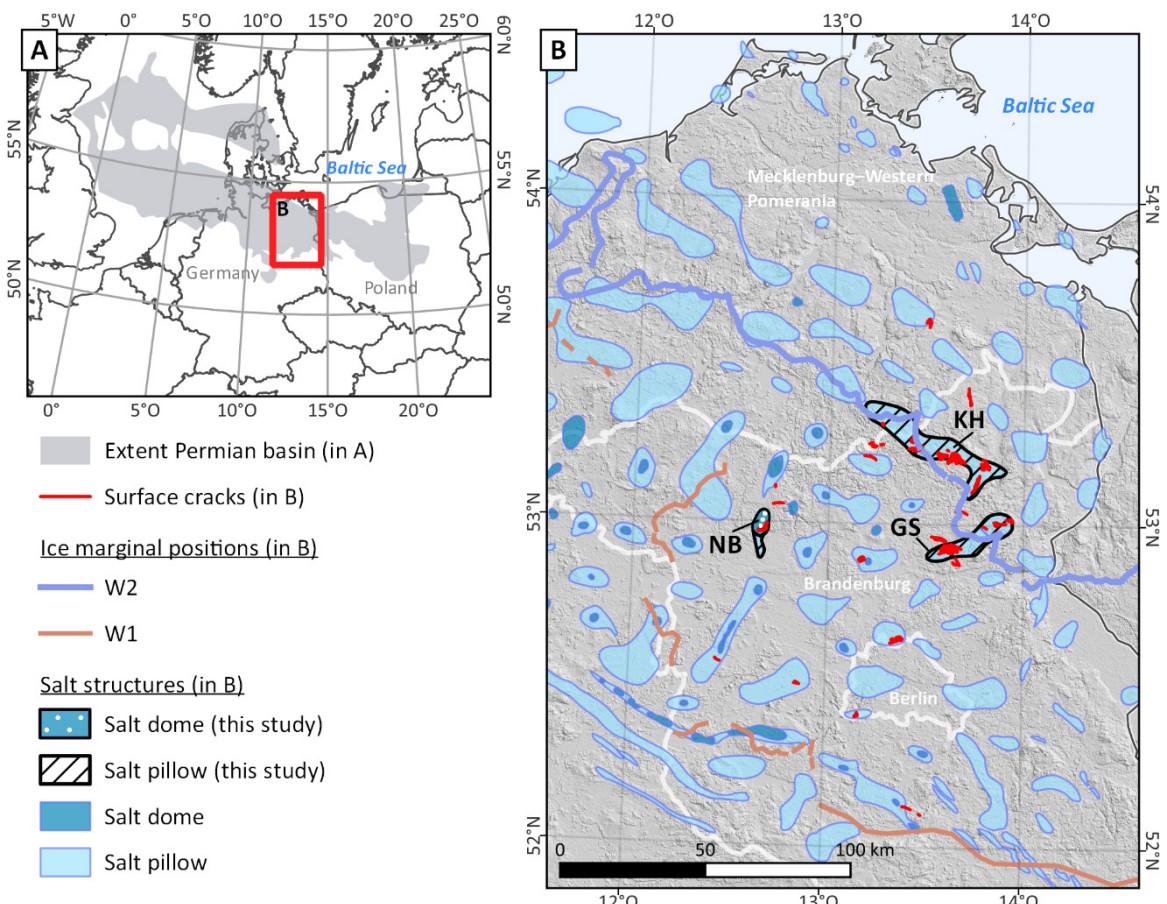

**Figure 1: Overview map. Bright blue polygons: salt pillows; dark blue polygons: salt domes (InSpEE, 2015). Polygons with black outline: Salt structures investigated in this study (see Figures 2 – 5 for detail maps). GS – Groß Schönebeck study site; KH – Klaushagen study site; NB – Netzeband study site. Brown line: LLGM (W1) ice extent, blue line: gLGM (W2) ice extent (Lüthgens and Hardt, 2022; Lüthgens et al., 2020). White lines in B are German administrative borders plotted for orientation.**

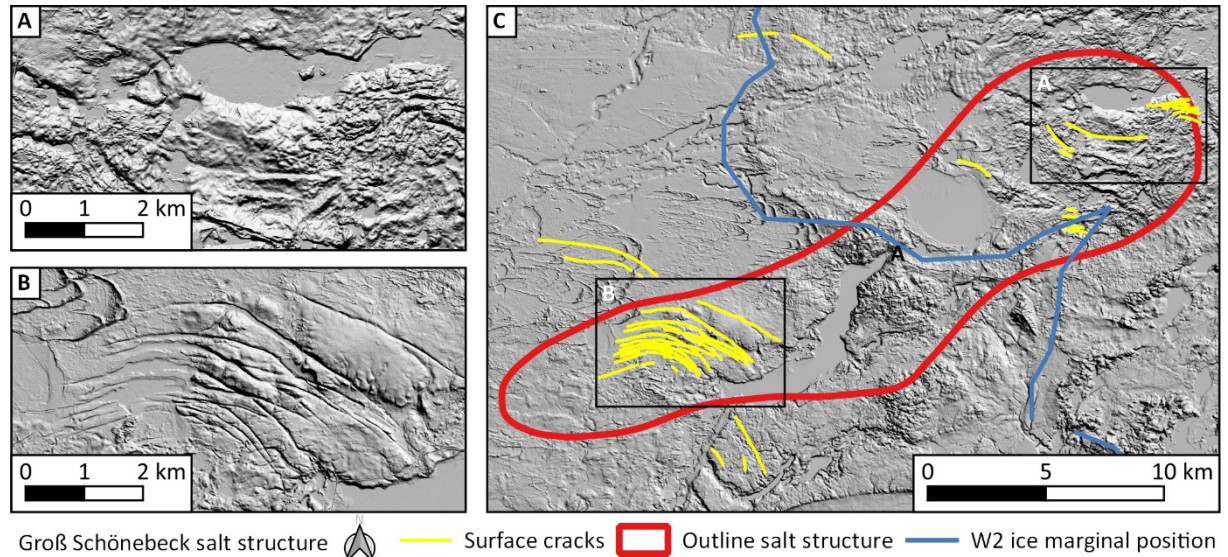

**Figure 2: Hillshade maps of the Groß Schönebeck salt pillow region. A: Detail view of the surface cracks found at the land surface atop the salt pillow in the Altkünkendorf area. B: Detail view of the surface cracks found at the land surface atop the salt pillow in the Schorfheide area. C: Location of the surface cracks with relation to the outline of the buried salt structure. Hillshade with 5x vertical exaggeration based on 1 m LiDAR digital terrain model (GeoBasis-DE/LGB, 2020).**

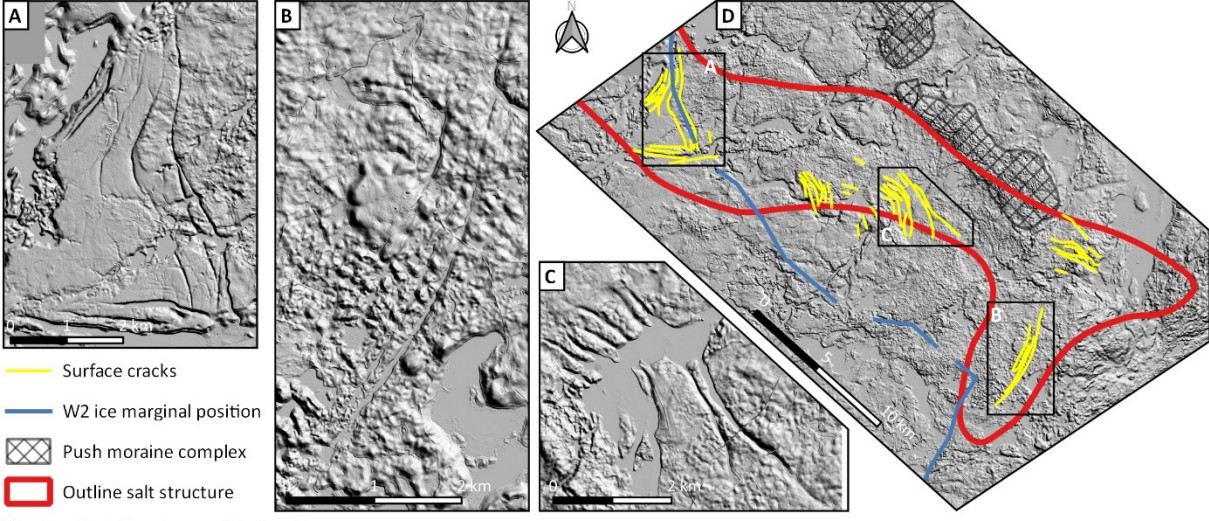

**Figure 3: Hillshade maps of the Triepkendorf-Klaushagen-Flieth salt structure region. A: Detail view of the surface cracks found at the land surface atop the salt pillow in the Thomsdorf area. B: Detail view of the surface cracks found at the land surface atop the salt pillow in the Friedrichswalde area. C: Detail view of the surface cracks found at the land surface atop the salt pillow in the lake Kuhzer See area. D: Location of the surface cracks with relation to the outline of the salt structure. Hillshade with 5x vertical exaggeration based on 1 m LiDAR digital terrain model (GeoBasis-DE/LGB, 2020).**

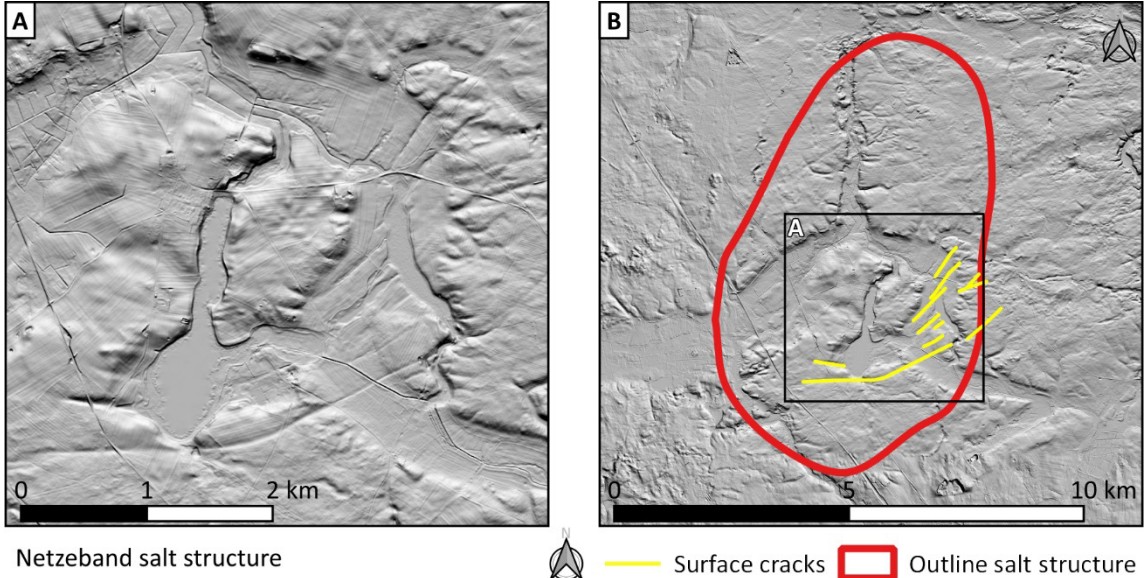

**Figure 4: Hillshade maps of the Netzeband salt dome region. A: Detail view of the surface cracks found at the land surface atop the salt dome. B: Location of the surface cracks with relation to the outline of the salt structure. Hillshade with 5x vertical exaggeration based on 1 m LiDAR digital terrain model (GeoBasis-DE/LGB, 2020).**

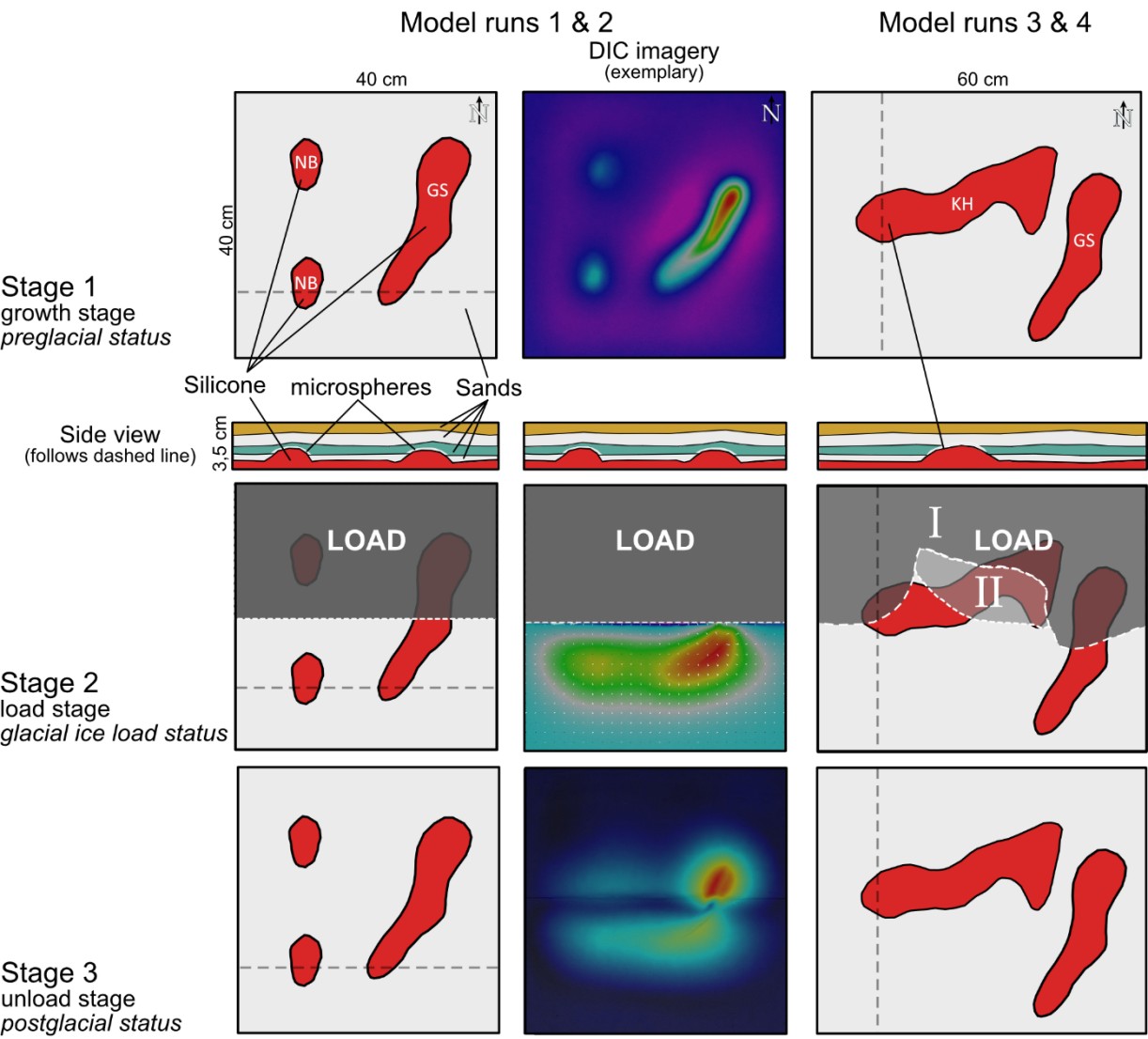

**Figure 5: Sketch illustrating the general physical model setup and the three model stages. The left column depicts the setup of model runs 1 and 2; the middle column shows exemplary DIC imagery of these runs. The right column depicts the setup of model runs 3 and 4, where the two domes were replaced by the KH salt pillow, which was parallel to the load margin. The middle image of the right column shows the load margin: In model run 4, first the areas "I" and "II" were loaded for 24 h, then area "II" was unloaded and area "I" was kept under load for another 24 h. Light grey dashed line in left and right columns depicts orientation of sections (Fig. 8).**

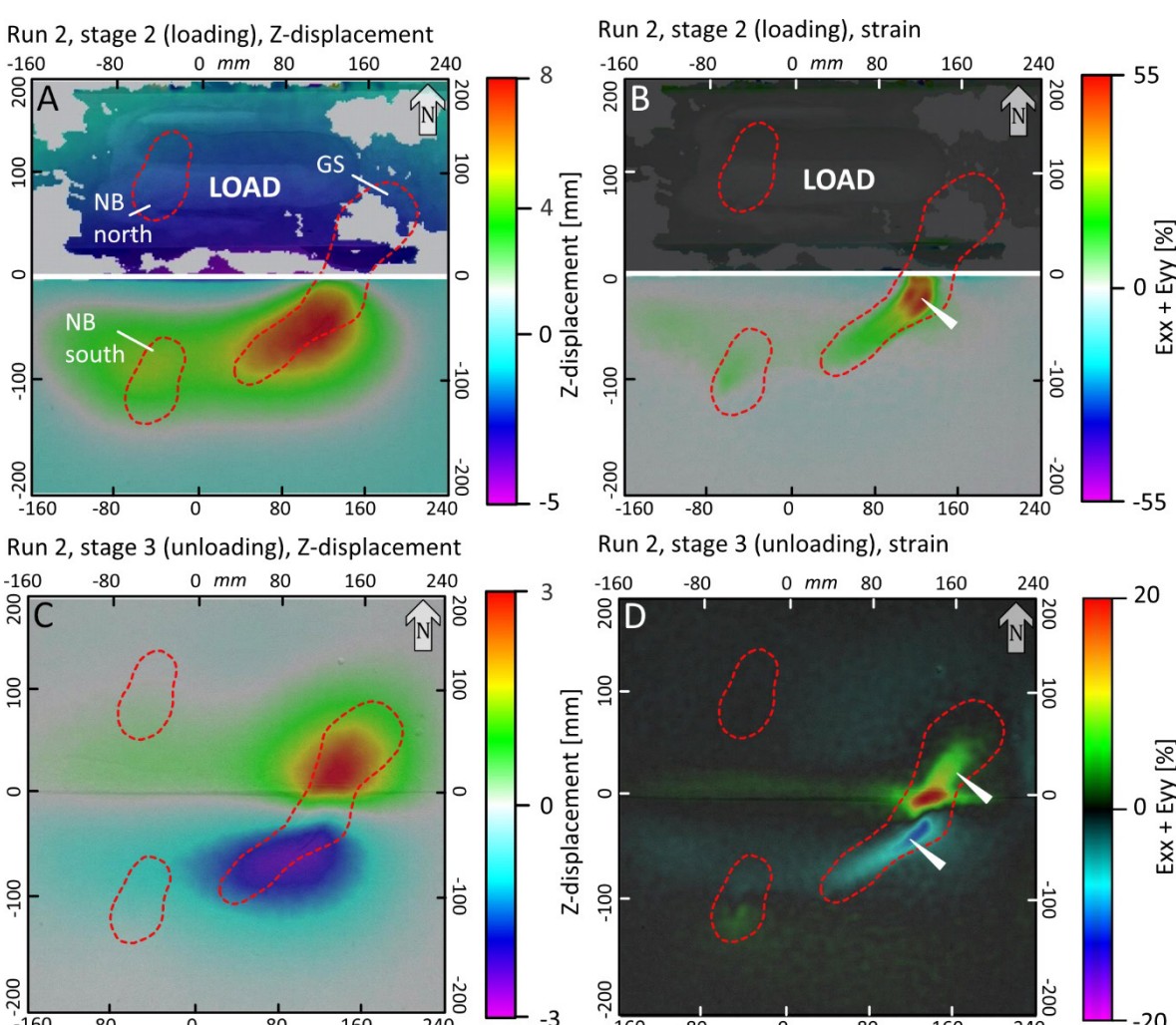

**Figure 6: Summarizing DIC imagery of run 2. In stage 2, the load was applied to the north of the horizontal white line. A: Z-map showing total vertical displacement in mm of stage 2. The grey colors in the upper half of the figure are "no data" areas. B: Strain [%] map of the total strain of stage 2. C: Z-map showing total vertical displacement in mm of stage 3. D: Strain [%] map of the total strain of stage 3. Red dashed outlines depict approximate position of salt structures. White arrows indicate position of crestal graben structure.**

675

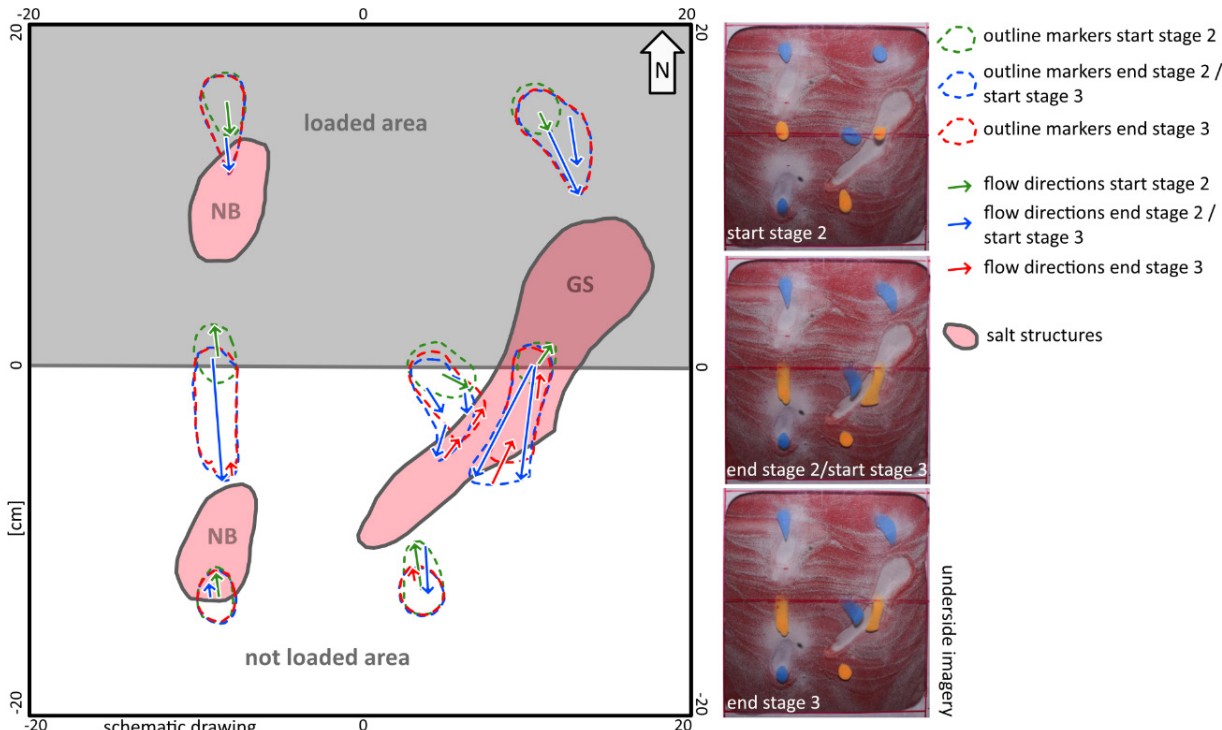

**Figure 7: Schematic drawing of flow patterns during three stages, based on underside imagery of run 2.**

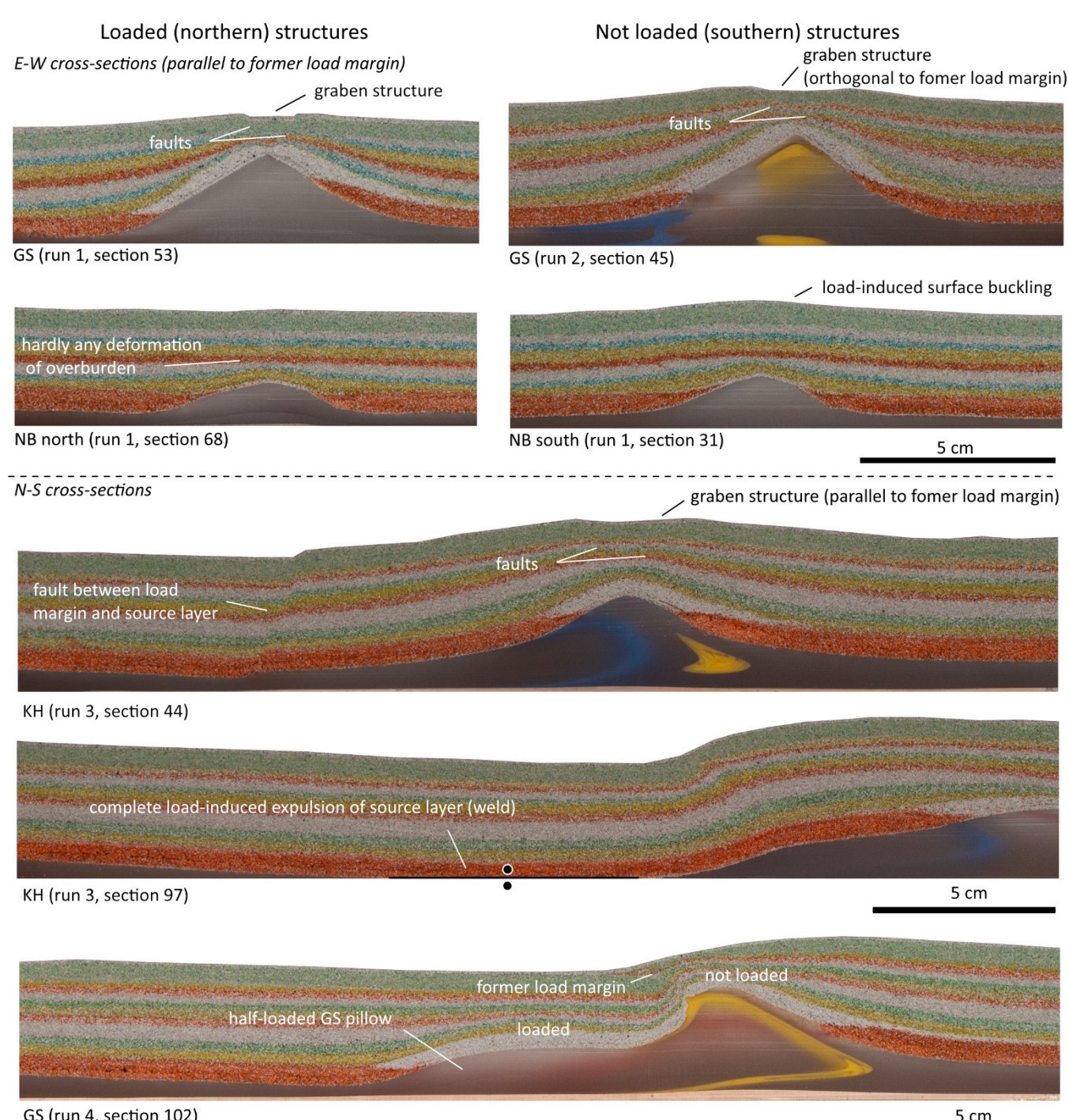

**Loaded (northern) structures**

*E-W cross-sections (parallel to former load margin)*

graben structure

faults

GS (run 1, section 53)

hardly any deformation of overburden

NB north (run 1, section 68)

**Not loaded (southern) structures**

graben structure (orthogonal to fomer load margin)

faults

GS (run 2, section 45)

load-induced surface buckling

NB south (run 1, section 31)

5 cm

*N-S cross-sections*

graben structure (parallel to fomer load margin)

faults

fault between load margin and source layer

KH (run 3, section 44)

complete load-induced expulsion of source layer (weld)

KH (run 3, section 97)

5 cm

former load margin    not loaded

half-loaded GS pillow    loaded

GS (run 4, section 102)

5 cm


**Figure 8: Examples of cross-sections from different model runs.**

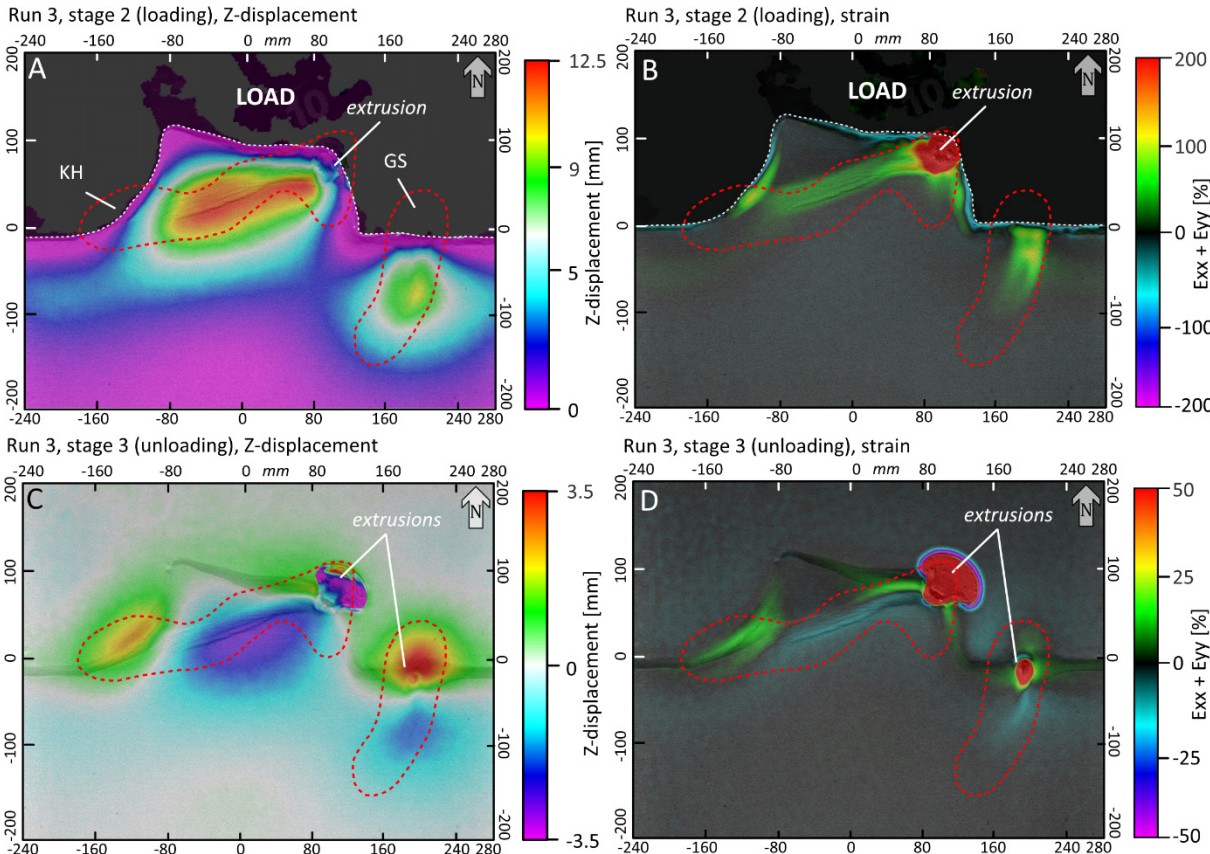

**Figure 9: Summarizing DIC imagery of run 3. In stage 2, the load was applied to the north of the dotted white line. A: Z-map showing total vertical displacement in mm of stage 2. B: Strain [%] map of the total strain of stage 2. C: Z-map showing total vertical displacement in mm of stage 3. D: Strain [%] map of the total strain of stage 3. Red dashed outlines depict approximate position of salt structures.**


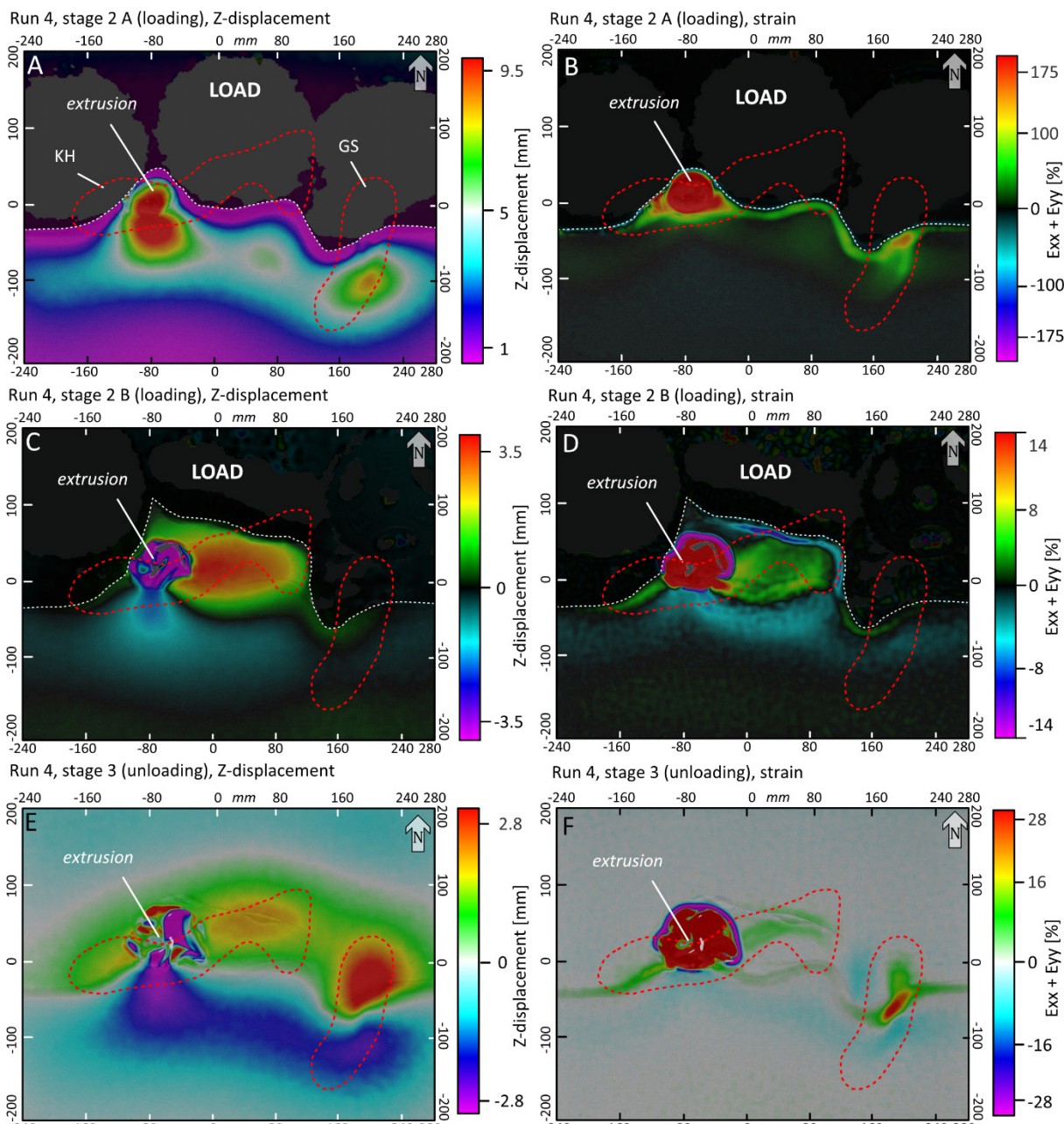

**Figure 10: Summarizing DIC imagery of run 4. In stage 2 A and 2 B, the load was applied to the north of the dotted white line. A: Z-map showing total vertical displacement in mm of stage 2 A. B: Strain [%] map of the total strain of stage 2 A. C: Z-map showing total vertical displacement in mm of stage 2 B. D: Strain [%] map of the total strain of stage 2 B. E: Z-map showing total vertical displacement in mm of stage 3. F: Strain [%] map of the total strain of stage 3. Red dashed outlines depict approximate position of salt structures.**


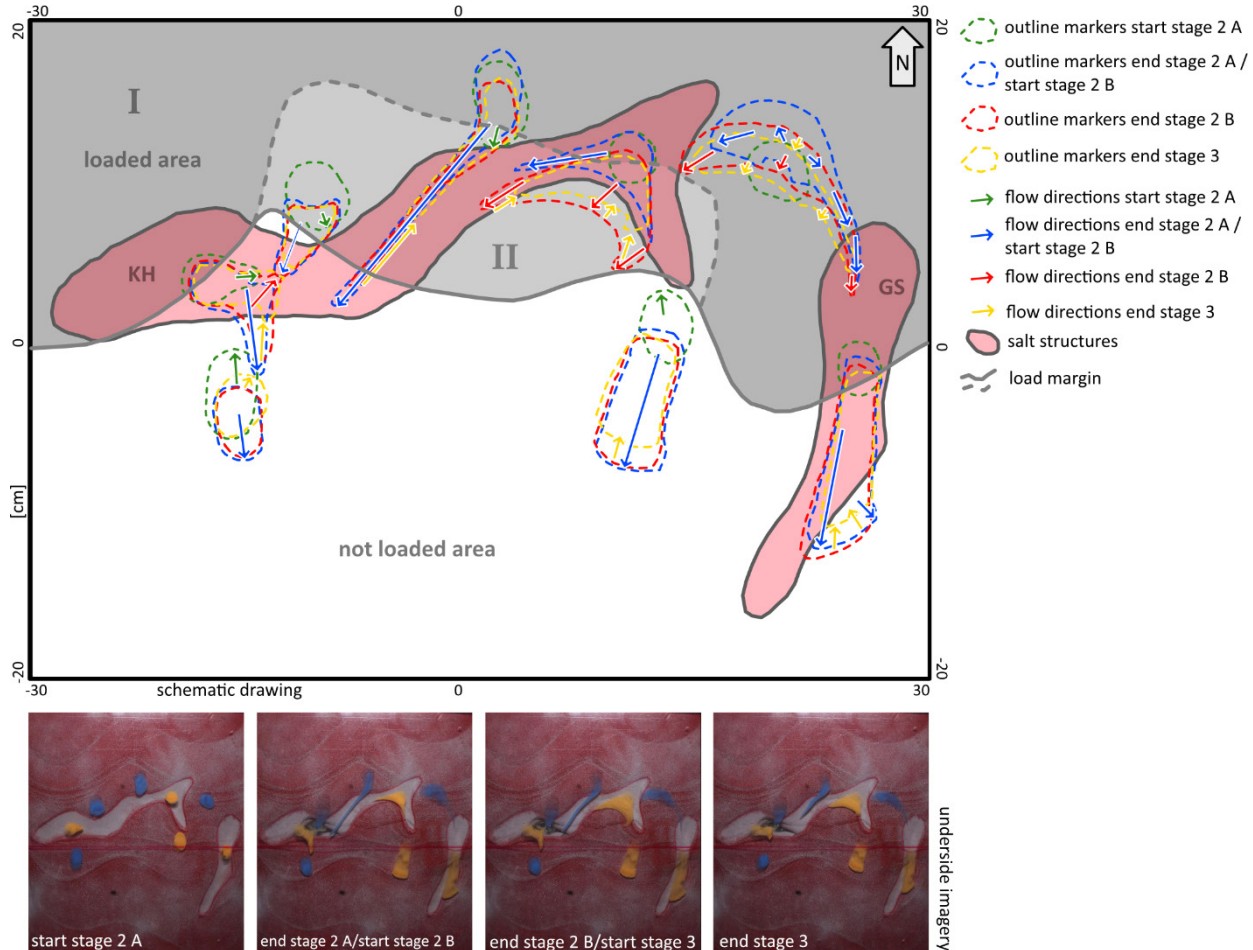

**Figure 11: Schematic drawing of flow patterns during three stages, based on underside imagery of run 4.**

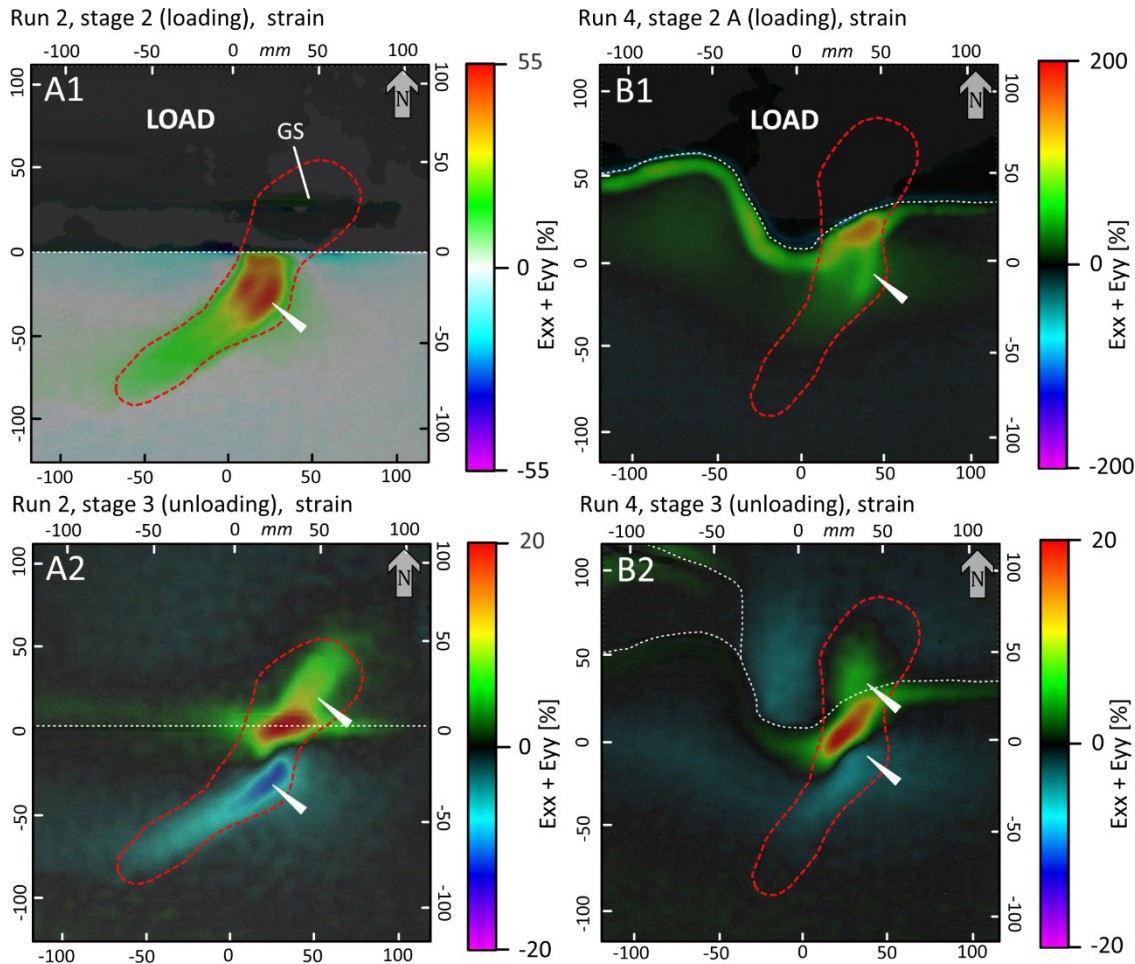

**Figure 12: Comparison of strain patterns above GS pillow using different load geometries (white dashed line): Left column - straight load margin; right column: lobate load margin. Red dashed outlines depict approximate position of salt structures. White arrows indicate position of crestal graben structures.**


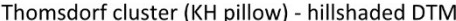

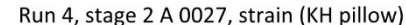

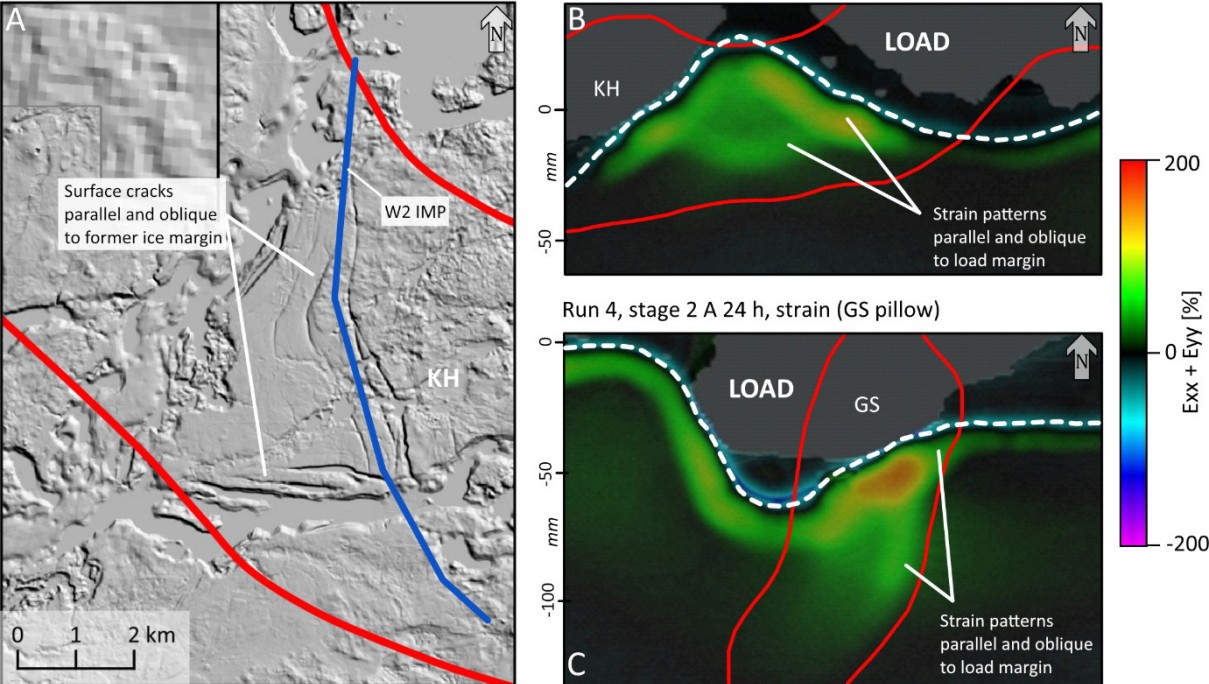

**Figure 13: Comparing surface cracks of the Thomsdorf cluster (A) at the KH salt pillow with strain patterns of stage 2 A in run 4 at the KH pillow (B) and the GS pillow (C) in relation to the ice/load margin. Both, the surface cracks and the strain maps, show patterns that are parallel and orthogonal in relation to the ice margin/load margin. Red dashed outlines depict approximate position of salt structures. The blue line in A depicts the approximate course of the W2 ice marginal position. White dashed line in B and C depicts the load margin.**

**Table 1: Natural and laboratory properties of salt structures investigated in this study.**

| | Groß Schönebeck (GS) | Netzeband (NB) | Triepkendorf-Klaushagen-Flieth (KH) |
|---|---|---|---|
| **Nature** | | | |
| **Type (geometry)** | Pillow ("J"-shaped saddle with three peaks) | Dome ("mushroom"-shaped dome connected to an elongated pillow) | Pillow (elongated "fish"-like shape) |
| **Size (length, width)** | 30 km, 8 km | 9 km, 6 km | 50 km, 6 – 18 km |
| **Orientation** | ENE-WSW | ESE-WNW | NNO-SSW |
| **Depth of top salt (asl)** | > -2300 m | > -500 m | > -2200 |
| **Relation to Weichselian ice dynamics** | Transgressed by W1 (lLGM) advance, half transgressed by W2 (gLGM) advance | Transgressed by W1 (lLGM) advance, not by W2 (gLGM) advance | Transgressed by W1 (lLGM) advance, half transgressed by W2 (gLGM) advance, close to W2 recessional push moraines (Angermünder Staffel) |
| **Relation to ice margin** | Perpendicular to ice margin | ./. | Parallel to ice margin |
| **Surface cracks** | More than 40, to the south and north of the gLGM ice margin, near to the southern and northern peaks of the salt structure | 10, to the east and south, approximate concentric arrangement with respect to the dome | 6 clusters, more than 50 cracks |
| **Laboratory (sandbox)** | | | |
| **Used in model run** | 1, 2, 3, 4 | 1, 2 | 3, 4 |
| **Size (length, width)** | 22 cm, 6 cm | 9 cm, 5 cm | 30 cm, 12 cm |
| **Relation to load** | Northern half loaded, southern half not loaded | Shape used twice in model: northern entity fully loaded, southern entity not loaded | Complex partially loaded pattern with lobate load margin and partial unloading in run 4 |

**Table 2: Model runs and selected parameters.**

| Run | Size [cm] | Salt structures | Source layer thickness | Loading phase (stage 2) / Unloading phase (stage 3) [h] | Load margin | Load weight |
|---|---|---|---|---|---|---|
| 1 | 40*40 | 1 pillow (GS) 2 domes (NB) | 8 mm | 48 / 48 | Straight | 20 lbs (9.07 kg) |
| 2 | 40*40 | 1 pillow (GS) 2 domes (NB) | 8 mm | 48 / 48 | Straight | 20 lbs (9.07 kg) |
| 3 | 40*60 | 2 pillows (GS + KH) | 8 mm | 48 / 48 | Lobate | 30 lbs (13.6 kg) |
| 4 | 40*60 | 2 pillows (GS + KH) | 8 mm | 24 / 24 / 96 | Lobate, with 2-phase unloading | 20 lbs (9.07 kg) |

**Table 3: Summary of maximum recorded vertical displacement rates. Runs 3 and 4 were excluded from mean calculation as the extrusions distort the result.**

| Run | Maximum DIC recorded vertical displacement [mm] | | | Difference [%] between stage 2 and 3 | Remarks |
|---|---|---|---|---|---|
| | End Stage 2 A | End Stage 2 B | End Stage 3 | | |
| 1 | 9 | | 4.75 | 53 | |
| 2 | 7 | | 2.8 | 40 | |
| Average of runs 1 and 2 | | | | **47** | |
| 3 | 12.5 | | 3.5 | 28 | Extrusion |
| 4 | 10 (24 h) | 4 (24 h) | 2.5 | 18 | Extrusion |