# Peer review of "Physical modeling of ice-sheet-induced salt movements using the example of northern Germany"

_EGUsphere, 2023_

## Author Comment (AC1)

**Reviewer #1**

Dear Jörg Lang,

We are grateful for your thorough and constructive comments. We appreciate your suggestions a lot. Thank you for helping us to improve our work.

Please find our replies to all your comments in the tables below.

Kind regards,

Jacob Hardt et al.

General comments

| Reviewer | Authors' response |
|---|---|
| Define goals / research questions need to be better defined. At the beginning, several questions are posed and the authors get back to those questions in the discussion. However, there is a mismatch between the questions posed at the beginning and those answered in the discussion and there are actually two separate sets of questions. My suggestion is to better define the main questions and placing the "secondary" questions within short explanations provided with each main question. | Thank you for detecting the mismatch between the questions in the intro and in the discussion! We have corrected that. Also, we followed your idea to merge the different sets of questions as follows:

*"1. What happens if salt structures are only partly covered by the ice and which role does the type of salt structure play?*
*This requires an investigation of different shapes and sizes of the salt structures during loading- and unloading processes.*

*2. What influence do the geometries of both ice margin and subsurface salt structure have on salt flow patterns?*
*This requires us to investigate loading- and unloading-induced intrasalt flow patterns.*

*3. Can these results help us to understand spatial patterns of present-day geomorphological features, such as surface cracks, above salt structures in northern Germany?"* |
| The issue of young salt tectonic activity in the study area should be explained in more detail. A challenge when studying ice-load induced salt movements is how to distinguish them from other (longer-term) movements. | Thank you! We rewrote the study area section significantly and also point to some of the sites where young salt activity was mentioned in the literature (Sperenberg, Rambow, Rüdersdorf). We also added this remark:

*"In general, it is challenging to differentiate ice-sheet induced salt tectonic movements from other (longer-term) tectonic movements, as the salt structures are usually coupled to tectonic lineaments (see discussion in Hardt et al., 2021). Therefore, approaches that take into account both the geomorphology and the deeper subsurface are necessary."* |
| The description of the study area should provide a bit more information | We provided a bit more context on the evolution of salt structures in the study area. |

| | |
|---|---|
| on the phases of salt tectonic activity. Have those salt structures been rising during the latest Cenozoic prior to the Pleistocene glaciations? Does it matter? | There is some sparse information that few salt structures in the study area were reactivated due to GIA, which is now also mentioned in the text (Rüdersdorf, Sperenberg, Rambow; Ludwig & Stackebrandt 2010). However, these are not the structures that we modelled in this study. In general, I think this is an interesting aspect that we should keep in mind for future investigations. At this point, we can't say, if it matters whether structures were active during the latest Cenozoic in the study area, or not. |
| Ice-sheet load in the models is not dynamically scaled (e. g., Lines 130 / 185ff). I am aware that scaling is a huge challenge for any physical model. However, the displacements in the models are very high in relation to the total thickness of the model section (>10 mm vertical displacement vs 35 mm thick model section). Salt extrusion observed in one run seems another example of extreme deformation. However, if the models are to be compared to natural examples and numerical models (the numerical models may of course have over shortfalls), somewhat more context should be provided and the limitations should be pointed out more clearly. Maybe a comparison to other physical models of salt tectonics helps to provide more context. I think the discussion would benefit from some more consideration here. | To increase the transparency on the issue of scaling, we have added a new section to the discussion titled "remarks on the scaling of our models". We acknowledge that the displacements are high with respect to the natural prototypes. We hope that the limitations of our models and our main intentions of studying three-dimensional processes rather than absolute rates are being more clearly communicated in the revised version of the MS. |

Detailed comments

| Reviewer | Authors' response |
|---|---|
| Line 11: Throughout the manuscript, you are switching between "glacial loading" and "ice-sheet loading / ice loading". Please check and decide for one term – I think "ice-sheet loading" is the most appropriate and widely used term | Thanks, we now consistently use the term "ice-sheet loading". |
| Line 12: This sentence is a bit misleading and should be split up and modified. The presented model is not only based on the Scandinavian ice sheet, but on the overall geological situation (salt structures, ice margins, etc.) in northern Germany. | Agreed, we have modified the sentence according to your suggestion: *"We investigate the influence of ice-sheet loading and unloading on subsurface salt structures using physical models, based on the geological setting of northern Germany, which* |

| | |
|---|---|
| | *was repeatedly glaciated by the Scandinavian Ice Sheet during the Pleistocene."* |
| Line 18: "salt-source layer" is the more common term. | OK – changed. |
| Line 28: Throughout the manuscript, you are switching between "salt structure", "salt dome" and "salt diapir". Please check for consistency and if the terms are used according to their definitions. | The different terms were chosen intentionally to differentiate between the different salt structure types (domes and pillows) that we modeled. We double checked the manuscript for consistency and included a definition for both terms, in agreement with a comment from Reviewer 2. |
| Line 35: The surface cracks are only one feature your models can help explain. I recommend making a broader statement here. | Thank you, we expanded it:

*"Our results lead to a better understanding of spatial patterns of the surface cracks that were mapped at the surface above salt structures and offer further room for interpretation of the influence of salt movements on the present-day landscape."* |
| Line 48: You should explicitly state here that future glaciations are considered a real issue for such long-term safety considerations. | Thank you, we added the aspect of future glaciations.

*"With regard to radioactive wastes, the long-term stability needs to be predicted for up to 1 Ma, and future glaciations are a factor that has to be considered in this case (BGE, 2020; Fischer et al., 2015)."* |
| Lines 50-51: Please rephrase or extent to explain the impact. | Thank you, we have sharpened the wording:

*"These salt tectonic processes can be triggered by large scale tectonic movements and changing sedimentary loads, which might be a result of changing climatic conditions including glaciations."* |
| Line 52ff: As glacio-isostatic adjustment and it's effects are important processes in the context of the study, this should be explained a bit better. Also, what are "hydrogeological adaptations"? Please elaborate. | OK, we have rephrased the passage and gave some more context. It now reads:

*"The load of the large Pleistocene ice-sheets pushed down the Earth's crust. In reaction to the unloading (i.e., melting of the ice-sheets), glacial isostatic adjustment (Lambeck et al., 2014) and processes such as postglacial rebound (Spada, 2017) set in and are still ongoing. As an example of postglacial rebound, Fennoscandia is still moving upwards, whereas regions to the south of the Baltic Sea (such as the study area) are moving downwards (Bungum & Eldholm, 2022). In addition, the ice-sheet advances modified subsurface hydro-thermal systems, which are still in the process of adapting to present-day conditions (Amberg et al., 2022; Frick et al., 2022)."* |

| | |
|---|---|
| Lines 57-59: This is misleading and oversimplifying: The link between neotectonics and ice loading is not just based on the parallel orientation. This would be a very weak link… | Thank you! This misleading sentence was also mentioned by Reviewer #3. I rewrote it accordingly. It now simply reads:

*"In northern Central Europe, postglacial seismic activity has been identified at several preexisting faults (Brandes et al., 2015; Müller et al., 2021)."* |
| Line 61: "the spatial distribution of" can be deleted. | OK – deleted it. |
| Lines 62 / 70: Again, there is a switch in terminology: Please use either "Zechstein salt" or "Permian salt". Permian may be more correct, as some salt structures in northern Germany may also include some Rotliegend salt. However, for your study area, Zechstein seems appropriate. | OK – we now consistently use "Zechstein salt". |
| Line 85ff: As those questions are central to the study, maybe use bullet points or number here to make them more striking. Also, there are only 3 questions here, while the discussion tries to answer 4 questions. It is a bit confusing that 3 questions are presented here and 4 similar, but slightly different questions in the next section. Maybe restructure this section, presenting higher-order questions (I think this is the second set) first, each question followed by a short explanation that may include the lower-order, more detailed questions (don't forget to modify the questions in the discussion accordingly!) Furthermore, all those questions (especially the first set of questions) should be put into a wider context, as your study is not just about checking some specific model configurations, but is a new approach to an understudied topic. | This goes together with your first general comment.
As outlined before, we are now using a list to highlight the questions. Also, we have merged the two sets of questions (which we think works fine) and added some more context. |
| Line 94-96: This sentence should be placed at the beginning of the next section. | We thought this sentence would be a nice transition into the next chapter – but we're absolutely fine with its new place at the beginning of the next section. |
| Line 102: Maybe rephrase to "…varies in thickness between…", as the thickness is the important part here. | Thank you, we changed it to:

*"The Mesozoic and Cenozoic overburden on the Zechstein salt varies in thickness in the region between more than 3000 m above deep-seated pillows, to only few hundred m above the highest salt domes (Stackebrandt and Beer, 2015) – some domes in northern Germany even pierce to the land surface (Künze et al., 2013; Sirocko et al., 2002; Stackebrandt, 2005)."* |

| | |
|---|---|
| Line 105: This is a huge leap from the very general features of the salt structures to the very specific surface cracks.
I wonder if the surface expressions of salt structures and their association with younger morphological features should be explained in more detail. | This whole section (study area) was significantly rewritten and expanded. We also rearranged this passage to make the shift from the general features to the local landforms more straightforward.
We wouldn't want to go into more detail on other morphological features here, as we want to keep the focus of the MS on the experiments and their discussion. |
| Line 110: I think we don't need a long description of the regional Quaternary geology here, but at least the term "W2" should be explained – please add just one sentence introducing the Weichselian ice advances into the area. | Agreed, we added two sentences on the Weichselian ice advances to the first paragraph of this chapter:

*"During the Weichselian, the study area was transgressed by the W1 advance, which occurred in late Maritime Isotope Stage 3. The Weichselian W2 advance occurred in Maritime Isotope Stage 2 and corresponds to the Last Glacial Maximum. The W2 advance reached only into the northern parts of the study area (Fig. 1; Lüthgens et al., 2020)."* |
| Line 155: Fault or strain pattern? | "Fault" is intended. When looking at the slabs, we were mainly interested in the faults. |
| Line 156ff: The first part of this section provides a lot of background information on the regional geology. I wonder if this should be better placed in the "Study area" – section, while the model set-up should focus on the model | We fully understand your concern. We've decided to keep the "study area" section more general and to provide the geological details that influenced how we precisely designed the models in the "model design" section. We think this way the paper is a bit more accessible to readers who are not so much into the study area but are more interested in the model design and the results, as they don't get lost by jumping back and forth between the chapters.

We would therefore advocate leaving the structure as it is. |
| Line 177: Please add: Was the model surface flattened? Were the sand layers compacted before loading? | Yes, the surface was flattened and no, the sand layers were not compacted before loading. Both statements were added to the respective section:

*"All subsequent layers of sand were added across the entire model without compacting them, just cresting our rising pillows and diapirs. In this way the model surface was flattened after each load was applied."* |
| Line 121: "ice-sheet load" seems more appropriate (see earlier comment). | OK. |
| Line 256: "Between W2 and W2"? Please check! May be rephrase to avoid the regional terminology. | This was indeed misleading. We clarified this passage: |

| | *"In model run 4, we used an even more undulating load margin as in run 3 for the first 24 h and then removed part of that load for the next 24 h, which is located between the W2 main ice marginal position and several recessional ice marginal positions (see Fig. 5 for illustration)."* |
|---|---|
| Line 280: Why place the cross sections in the supplement only? I recommend adding them as a regular figure. | We discussed this a lot and initially decided to move them to the supplement, as the results are mainly based on the strain and Z maps and as there are a lot of figures already. However, we are happy to follow your recommendation and brought the cross sections back into the main text (new Fig. 8), adding explanations to the respective results sections. |
| Line 281: Discussion: Please see my earlier comments on the questions. If you decide to modify the questions / goals, some reorganization may be necessary here. However, I don't see any issues with the overall structuring of the discussion with the questions as section headings. | Thank you, we took care of that with respect to your other comments! |
| Line 288, 300: Is "connectivity" the appropriate term here? Please consider rephrasing. | Thank you! We believe that this is the appropriate term for the process we aim to describe. |
| Lines 290-291: The last sentence of the section starts the interesting part of the discussion. Please elaborate further. | We have merged this section with the next one and do elaborate on this point in the section "Can these models help us…", which we have also slightly expanded. |
| Line 293: My impression is that the focus of this section is rather the position of the (ice) load margin relative to the salt structure than the type / size of the salt structure. Please also check with the next section. Maybe the section should better be combined. | Thank you. We have merged this section with the previous one and believe it reads better now. |
| Lines 294-295: Please define "larger" and "smaller". Does this refer to the area, volume of salt or else? | Here we refer to the size of the structures in plan view. We made this clearer in the text:

*"When comparing our different modeled structures in plan view, it is apparent that the bigger structures (in this case the pillows) showed significantly stronger reactions to the loading and unloading cycles than the smaller structures (in this case the domes), irrespective of roof thickness and strength."* |
| Lines 305-307: The resistance to salt flow caused by thinning salt layers is a well known phenomenon in salt tectonics. Some references seem appropriate here. | We slightly changed the wording and included appropriate references (Hudec and Jackson, 2007; Wagner & Jackson, 2011)

*"Consequently, the NB domes were placed at a distance from the load margin in our models. These structures lacked connectivity as they"* |

| | *were not part of a salt corridor at depth and were only fed via the thinned source layer, which was additionally confined by the rim synclines adjacent to the diapirs, which increase the resistance of salt to flow (Hudec & Jackson, 2007; Wagner & Jackson, 2011)."* |
|---|---|
| Line 349: This is not exactly what we wrote. Our point was rather that a relatively small obstacle may initiate the formation of glacitectonic thrusts. | We regret the little misunderstanding and corrected the passage accordingly: *"Lang et al. (2014) concluded that salt rise alone would not be sufficient to create obstacles large enough to stop an inland ice sheet, but suggested that rising salt structures in the foreland of an advancing ice-sheet may favor the formation of glacitectonic thrusts."* |
| Furthermore, I am still skeptical about rising salt structures acting as (significant) obstacles to ice flow, as the ice sheets did transgress other, even higher topographic obstacles, e.g. some low mountain ranges near the maximum extends of the Middle Pleistocene ice sheets in northern Germany. | To your second point: we can only say that this definitely needs more research. It is true that ice sheets are capable of transgressing high obstacles. But as you also mentioned in your 2014 paper, there are some (few) examples of obstacles acting like a nunatak (e.g., Sperenberg dome). I can imagine that at the edge of the ablation zone the topographic highs don't have to be too large to form an obstacle to the ice sheet - but this requires more dedicated investigations. |
| Line 361: I find your observation of the intense deformation in salt structures that are partly loaded very interesting. The reason we placed the ice-margin 1000 m away from the salt structure in the numerical models was exactly the strong deformation occurring if the ice margin was located exactly on top of the salt structure. This strong and rapid deformation commonly triggered the numerical models to crash. We never really addressed this issue in our papers. I fully agree that such a configuration should result in larger displacements. | Thank you, we appreciate your very interesting comment concerning your own model results. It is good to know that your models seemed to go in a similar direction when the structures were partly loaded! This is a promising aspect for future research! |
| Figure 5: in the central part of the figure, it looks like the orientation of the load is different in the cross section and in the map view. The map shows a left-right (west-east?) trending margin of the load. I would understand the cross-section to show a top-bottom (north-south) trending margin. Please clarify. | Thank you for pointing this out! You are right, the depiction of the weight and the metal plate in the second line in the center of the figure ("side view") is misleading. This was mistakenly adopted from an earlier version of this figure, where the map views were rotated 90 degrees to the left. I removed the depiction of the weight from the side view as we believe, this little extra detail is not necessary for the comprehension. (see below) |
| Figures 6, 8, 9 and 11: I suggest naming the stages above each panel, so the reader does not | Done! We added the terms "loading" or "unloading" above each panel. |

| have to look up what "stage 2" actually represents. | |
|---|---|
| Supplement: I wonder why the supplement (one figure showing cross sections from the model) is not included as a regular figure? You don't show any cross sections, so this might be a nice addition. | Solved with similar comment above. Sections are now a regular figure. |

Revised Figures

[Figure]

**Figure 1: Sketch illustrating the general physical model setup and the three model stages. The left column depicts the setup of model runs 1 and 2; the middle column shows exemplary DIC imagery of these runs. The right column depicts the setup of model runs 3 and 4, where the two domes were replaced by the KH salt pillow, which was parallel to the load margin. The middle image of the right column shows the load margin: In model run 4, first the areas "I" and "II" were loaded for 24 h, then area "II" was unloaded and area "I" was kept under load for another 24 h. Light grey dashed line in left and right columns depicts orientation of sections (Fig. 13).**

[Figure]

**Figure 2: Summarizing DIC imagery of run 2. In stage 2, the load was applied to the north of the horizontal white line. A: Z-map showing total vertical displacement in mm of stage 2. The grey colors in the upper half of the figure are "no data" areas. B: Strain [%] map of the total strain of stage 2. C: Z-map showing total vertical displacement in mm of stage 3. D: Strain [%] map of the total strain of stage 3. Red dashed outlines depict approximate position of salt structures. White arrows indicate position of crestal graben structure.**

[Figure]

**Figure 3: Summarizing DIC imagery of run 3. In stage 2, the load was applied to the north of the dotted white line. A: Z-map showing total vertical displacement in mm of stage 2. B: Strain [%] map of the total strain of stage 2. C: Z-map showing total vertical displacement in mm of stage 3. D: Strain [%] map of the total strain of stage 3. Red dashed outlines depict approximate position of salt structures.**

[Figure]

**Figure 4: Summarizing DIC imagery of run 4. In stage 2 A and 2 B, the load was applied to the north of the dotted white line. A: Z-map showing total vertical displacement in mm of stage 2 A. B: Strain [%] map of the total strain of stage 2 A. C: Z-map showing total vertical displacement in mm of stage 2 B. D: Strain [%] map of the total strain of stage 2 B. E: Z-map showing total vertical displacement in mm of stage 3. F: Strain [%] map of the total strain of stage 3. Red dashed outlines depict approximate position of salt structures.**

[Figure]

**Figure 5: Comparison of strain patterns above GS pillow using different load geometries (white dashed line): Left column - straight load margin; right column: lobate load margin. Red dashed outlines depict approximate position of salt structures. White arrows indicate position of crestal graben structures.**

---

## Author Comment (AC2)

**Reviewer #2**

Dear Chris Clark,

Thank you very much for reviewing our manuscript and for your suggestions on improvements. We are very happy about your positive assessment of our work. The following table lists our responses to your comments.

Kind regards,

Jacob Hardt et al.

| Reviewer | Authors' response |
|---|---|
| Yes you say the applied load is unrealistically high and is mostly to speed up the experiments which seems fine. But maybe you could tell us by what rough factor; take a typical low parabolic surface profile from an ice sheet margin and estimate the likely ice loading. | Yes, the load is several orders of magnitude higher than it would be in the real world. From the start of the planning of the experiments, we decided not to scale the load, as this would have introduced an unknown number of new uncertainties to the models. We wanted to solely concentrate on the processes, not on absolute rates. We think it is a pretty complex endeavour to model the ice volume near the LGM ice margin in northern Germany. However, we think there are ways to do it, but it will require a lot of thought and dedication. Now that we have the "base work" done, we feel confident enough to introduce more complexity to our future models on this subject – and one aspect is for sure finding useful ways for a proper scaling of the ice-sheet load. In the revised version, we emphasize even more that the applied load was drastically exaggerated, but we would like to refrain from going into details on how much – simply because it was not part of the model design and because we do not want to give room for any misinterpretations that could arise from the indication of an order of magnitude. |
| The percentage z movements appear large compared to the dimensions of the 'sediments' in the sandbox. You tell us that this is likely unrealistic which is why you don't fully explore the absolute Z movements, but could you say more on this; the extent to which they likely arise from viscosity and loading scaling issues vs, static ice margins against mobile ones which might not have enough elapsed time for bulges to develop. | We have added a new section to the discussion section that addresses the issue of the loading scaling ("remarks on the scaling of our models"). We will have to examine a dynamically scaled load in future experiments – it was outside the scope of our initial models and their possible impacts are hard to elucidate on basis of our current knowledge. |
| It would help to tell the reader early in the paper what you actually mean by pillow vs domes, later on I gathered it was combination of factors relating to size and | Thanks for this remark, we added a brief definition in the section "study area": *"The main structure types in the NE German sector of the CEBS are salt pillows, although several* |

| | |
|---|---|
| depth, but am still a bit unclear, and yet you often distinguish between them. | *domes exist, too. Salt pillows have a parallel contact with suprasalt strata, whereas salt domes have discordant contacts with their upper strata (Jackson & Hudec, 2017b).”* |
| Line 195  stage 1 growth. Suggest to make clearer in wording that this stage is a 'relaxing' stage such that the materials initialise prior to your experiment. Use of the word 'growth' first implied to me that stuff was responding to an advancing ice load. | Agreed! We now termed it *“initial growth stage”* and added as first sentence: *“Preparation of the model environment before start of the actual experiments”* |
| Explain what  the white lines are in fig 1b | OK, added to the figure caption: *“White lines in B are German administrative borders plotted for orientation.”* |
| The salt dome stipple is not very visible in fig 1b | Thank you! I changed the stipple color to white in order to improve visibility (see below). |
| Explain in fig caption what  the grey is in fig6A? no data or a value I cant see in colour bar? | Thanks! We added to the caption that these are *“no data”* areas. |

Revised Figure 1

[Figure]

**Figure 1: Overview map. Bright blue polygons: salt pillows; dark blue polygons: salt domes (InSpEE, 2015). Polygons with black outline: Salt structures investigated in this study (see Figures 2 – 5 for detail maps). GS – Groß Schönebeck study site; KH – Klaushagen study site; NB – Netzeband study site. Brown line: LLGM (W1) ice extent, blue line: gLGM (W2) ice extent (Lüthgens and Hardt, 2022; Lüthgens et al., 2020). White lines in B are German administrative borders plotted for orientation.**

---

## Author Comment (AC3)

**Reviewer #3**

Dear Peter B.E. Sandersen,

We thank you for reviewing our manuscript and for the constructive comments you gave. We appreciate your efforts a lot, as they helped us to improve our work. The following tables list our responses to your comments.

Kind regards,

Jacob Hardt et al.

General comments:

| Reviewer | Authors' response |
|---|---|
| In the Introduction it is mentioned (lines 56 to 59) that in areas affected by GIA fault zones can be reactivated ('glacially induced faults'). However, this is the only place in the paper where this is mentioned. Obviously, it is not in the scope of the laboratory modelling to include the effect on the upper mantle, but I suggest that a sentence or two are added to the Discussion, where it is discussed whether/to which extent this mechanism could have affected the area chosen as study area. | We agree that ice-sheet induced salt movements and glacially induced faults (GIF) at one point deserve an integrative consideration. By mentioning the GIFs in the introduction, we want to transport the message to readers unfamiliar with the region that the ice advances are capable of triggering deep movements and that the salt movements that we investigate are one possible additional component triggered by the ice.
While thinking about your comment, we did not find an appropriate spot in the discussion where to briefly discuss our results in the context of GIFs, especially as this was out of the scope of our experiments.
We believe this requires more work in a dedicated study. |
| The 'Experimental methodology and setup' section starts with 'Remarks on the selection of model parameters' followed by 'Modeling materials and data capture' and finally 'Model design'. I suggest that the 'Remarks' section be moved to the end because the reader cannot necessarily relate to remarks on the individual stages of the modelling before the model has been described. Also, consider moving the sentences of lines 185 to 193 to the end of the 'Remarks' section as it, in my opinion, fits better here. | Thank you! We agree and followed your advice! |
| In the 'Results' section you mention the crestal grabens that form above the modelled salt structures. As the figures generally are small, please refer more specifically to where on the figures the crestal grabens can be seen (i.e. with arrows). | Done! We've added white arrows to the respective images and added a note on that in the text and in the figure captions. (see below) |

| As the timescales of salt flow and ice flow are very different, I agree that it is obvious that the loading in the modelling can only be stationary. It is also understandable that the modelling cannot be weighted, and that the ice load in the model setup has to be exaggerated. However, when specifically evaluating the effect of a lobate ice margin, I feel - due to the factors mentioned – that the uncertainties on the model results here must be quite large. I suggest more elaboration on this in the discussion, for instance as a separate part of the discussion dealing with uncertainties. | This is an issue that was also pointed out by the other reviewers. As you suggested, we added a whole new section "remarks on the scaling of our models" to increase the transparency regarding the scaling. |
|---|---|
| It is mentioned as a result, that 'the reversed vertical displacement after the unloading, caused by the flow reversal of the salt system accounts for only up to roughly 50 % of the vertical displacement that occurred during the loading stage' (sentence from the Conclusions). But there are no suggestions as to why this is happening. Please elaborate on why the system does not return to the pre-loading situation but instead establishes a new equilibrium, and to which extent is it believed that the chosen model setup can be responsible for some of the observed differences (the static load, a non-weighted model, the extrusion etc.)? | Thank you, this is a very interesting remark. We don't think that the system had quite reached a new equilibrium after the end of the unloading stage. Although we made sure that the unloading stage was long enough and we didn't record any significant movements before we finished the models, we do believe that very slow processes of reequilibration would have continued for some time to come. This is an effect of the high body forces applied during the loading stages versus the low body forces during the unloading stages. Also, the back flow was favored by the high connectivity within the pillows, whereas the load resulted in a thinning of the source layer, which decreased the flow reversal capacity outside the structures.

This is a complex issue, which will require further work. We added a few lines on that in the new discussion section ("remarks on the scaling of our models"):

"*Although we witnessed a flow reversal during the unloading stages, the vertical displacement rates during the unloading stages only accounted for roughly 50% compared to those from the loading stages. This is most likely an effect of the very different body forces involved, which were high during the load stage and low during the unloading stage. In addition, the back flow was favored within the pillows, where the salt is thick and connectivity is high. Outside the pillows the source-layer salt was thinned during loading by expulsion into the pillows and diapirs, and thus the flow resistance through these thinner conduits increased for the unloading stage. This thinning and increased flow resistance impacted the process of reequilibration driven solely by gravity,* |

| |
|---|
| *which we would expect would to have continued very slowly for some time to come. The process of decreasing salt flow in thinning salt layers is well known from research into salt welds and has to do with specific salt viscosities and internal impurities within the salts (e.g., Wagner & Jackson, 2011; Jackson and Hudec, 2017a)."* |

Specific comments:

| Reviewer | Authors' response |
|---|---|
| Line 57-59: Please re-think this sentence and the argument it contains: The orientation of faults parallel to the Pleistocene ice margins does not document a link between neotectonic activity and ice sheet loading. | This was also mentioned by Reviewer #1 and the passage was rewritten accordingly:

*"In northern Central Europe, postglacial seismic activity has been identified at several preexisting faults (Brandes et al., 2015; Müller et al., 2021)."* |
| Line 71: Delete 'at the surface'. | OK. |
| Line 90: The loading/unloading processes are unrelated to the size/shape of the salt structure. Consider writing 'during the loading and unloading processes'. | Thanks, now it reads:

*"This requires an investigation of different shapes and sizes of the salt structures during loading- and unloading processes."* |
| Line 91: I would prefer to write 'explore the relation between'. | This whole part was rewritten following the suggestions of Reviewer #1. |
| Line 94: Consider 'Northern Germany constitute an ideal study area, as it is….' instead of 'Northern Germany provides the ideal model region for our study, as it is….' | OK:

*"Northern Germany constitutes an ideal study area, as it is rich in various types of subsurface salt structures, was repeatedly glaciated during the Pleistocene, and provides several areas where geomorphological landforms point to a salt tectonic influence."* |
| Line 103: With 'low-lying', do you mean 'deep-seated'? | Thanks, "deep-seated" is indeed the better term.

*"The Mesozoic and Cenozoic overburden on the Zechstein salt varies in thickness in the region between more than 3000 m above deep-seated pillows, to only few hundred m above the highest salt domes (Stackebrandt and Beer, 2015) – some domes in northern Germany even pierce to the land surface (Künze et al., 2013; Sirocko et al., 2002; Stackebrandt, 2005)."* |
| Line 107: Consider using 'terrain surface' instead of 'free surface' | The whole sentence was rewritten:

*"The so-called surface cracks are interpreted as expansion ruptures due to salt flow triggered by loading- und unloading effects of the SIS, which* |

| | *eventually resulted in upwards movement of pillows and domes.”* |
|---|---|
| Line 136: Write 'silicone flow' rather than 'salt flow'. | OK:

*"Powdered pigments were mixed with the silicone and added as passive markers to several locations in the source layer in order to track the silicone flow.”* |
| Line 159: Please explain what you mean with: 'The GS and KH have a heterogeneous geometry with several peaks'. | We clarified it and changed it to:

*"The GS and KH salt pillows have an undulating topography with several peaks.”* |
| Line 162: Should the sentence '…stimulating the debate of the relationship between salt structures and ice sheet extent' be moved to the discussion? In my opinion it is irrelevant here. | Agreed, we moved this passage to the discussion (section "can these models help us…") and slightly modified it:

*"Interestingly, the spatial correlation between salt pillows and the W2 ice marginal position has initially led to the development of the theory of a dynamic relationship between salt structures and the ice extent (Gripp, 1952; Schirrmeister, 1998) and our results revealed the largest deformations in comparable settings.”* |
| Line 166. Consider deleting '…thus providing a promising modeling scenario', because it is a subjective evaluation at this stage. If what you mean is that it would be interesting to model a scenario like this because the salt structures were partly transgressed, please rephrase. | Agreed, we rephrased it to:

*"[…] which provides a setting that corresponds with the focus of our research questions.”* |
| Line 179: 'Front edge' instead of 'leading edge'? | OK.

*"In the later runs, a metal plate with an undulated front edge was used to simulate the lobate nature of ice margins (Fig. 5).”* |
| Line 212: Consider reducing 'covered by the glacial load during the loading stage' to simply 'loaded'. | OK. |
| Line 282: 'Here, we will……….attempt to discuss…'. Delete 'attempt to'. | Agreed! |
| Line 325: 'Keeping the ice dynamics of the two different Weichselian ice advances……in mind,….' What is meant here apart from the spatial extent of the ice advance? | Thank you! I have deleted the term "ice dynamics" and only relate to the different ice extents now. Ice dynamics were not in the scope of this paper and shall be addressed in a different study.

*"Keeping the spatial extents of the two different Weichselian ice advances in mind (Fig. 1), the distribution of the surface cracks may be explained on basis of the results gained from our physical models.”* |

| Line 355: '….the advancing ice sheet would push an intrasalt 'bowwave' in front of it, giving rise to the structures in front of it': As modelling does not include the dynamics of the ice sheet and given the large differences in time scales of salt movement and ice-sheet movement, I find that this conclusion is difficult to make based on the modelling. | OK, we deleted this interpretation. It shall be addressed in future projects. |
|---|---|

Revised Figures

[Figure]

**Figure 1: Summarizing DIC imagery of run 2. In stage 2, the load was applied to the north of the horizontal white line. A: Z-map showing total vertical displacement in mm of stage 2. The grey colors in the upper half of the figure are "no data" areas.  B: Strain [%] map of the total strain of stage 2. C: Z-map showing total vertical displacement in mm of stage 3. D: Strain [%] map of the total strain of stage 3. Red dashed outlines depict approximate position of salt structures. White arrows indicate position of crestal graben structure.**

[Figure]

**Figure 2: Comparison of strain patterns above GS pillow using different load geometries (white dashed line): Left column - straight load margin; right column: lobate load margin. Red dashed outlines depict approximate position of salt structures. White arrows indicate position of crestal graben structures.**

---

## Author Response (AR2)

Dear Peter Sandersen,

Thanks again for taking your time to review our manuscript. We are very happy about your positive assessment of our revised work.

**R:** Line 133-1134: I relation to possible deeper tectonic movements, the authors write: 'Therefore, approaches that take into account both the geomorphology and the deeper subsurface are necessary'. I agree on this, but the deep subsurface part is not included in the modelling. Please consider a short remark on this.

**A:** Thank you for this suggestion. We feel that a remark on our models would not fit to the "study area" section, which you refer to. Thus, we added the following sentence to section 3.2 Model design (l. 170 f.):

"Deeper subsurface features beneath the salt structures were not included in the models."

**R:** Line 315: Consider moving this interpretation to the discussion.

**A:** OK, I deleted the statement from the text, as there is no suitable connection point in the discussion.

**R:** Line 354: '….highlighting the significance of the geometry of the load': Consider moving this interpretation to the discussion.

**A:** Thank you, I deleted this statement. This finding already has a lot of room in the discussion and doesn't need to be mentioned in the results.

**R:** Line 409: Consider using 'previously' instead of 'initially'.

**A:** Done, changed it to "previously".

**R:** Line 474: I am unsure what is meant with 'the very different body forces'. Consider clarifying.

**A:** We tried to clarify this by inserting short explanations (green) to the sentence in question:

"This is most likely an effect of the very different body forces involved, which were high due to the high applied load during the load stage and low (only gravity and sedimentary overburden) during the unloading stage."

**R:** Lines 473-480: Just a comment: When the salt movements create structurally damaging deformations of the sediments above, especially when extrusion occurs, I would not expect the system to fully return to the pre-loading situation. Please consider if this irreversibility to a certain degree could explain why the vertical displacement is higher during the loading compared to the unloading.

**A:** Thank you for your thoughts! Yes, the influence of structural deformations on the salt flow during the unload stage is another very interesting aspect of the system. We will continue to take this into account in future studies.

Kind regards,

Jacob Hardt